# SIMPLE: SPECIALIZED MODEL-SAMPLE MATCHING FOR DOMAIN GENERALIZATION

**Ziyue Li**[1],[*] **Kan Ren**[2], **Xinyang Jiang**[2], **Yifei Shen**[2], **Haipeng Zhang**[1], **Dongsheng Li**[2]
[1]ShanghaiTech University, [2]Microsoft Research
`litzy0619owned@gmail.com, kan.ren@microsoft.com`

## ABSTRACT

In domain generalization (DG), most existing methods aspire to fine-tune a specific pretrained model through novel DG algorithms. In this paper, we propose an alternative direction, i.e., to efficiently leverage a pool of pretrained models without fine-tuning. Through extensive empirical and theoretical evidence, we demonstrate that (1) pretrained models have possessed generalization to some extent while there is no single best pretrained model across all distribution shifts, and (2) out-of-distribution (OOD) generalization error depends on the *fitness* between the pretrained model and unseen test distributions. This analysis motivates us to incorporate diverse pretrained models and to dispatch the best matched models for each OOD sample by means of recommendation techniques. To this end, we propose SIMPLE, a specialized model-sample matching method for domain generalization. First, the predictions of pretrained models are adapted to the target domain by a linear label space transformation. A matching network aware of model specialty is then proposed to dynamically recommend proper pretrained models to predict each test sample. The experiments on DomainBed show that our method achieves significant performance improvements (up to 12.2% for individual dataset and 3.9% on average) compared to state-of-the-art (SOTA) methods and further achieves 6.1% gain via enlarging the pretrained model pool. Moreover, our method is highly efficient and achieves more than $1000\times$ training speedup compared to the conventional DG methods with fine-tuning a pretrained model. Code and supplemental materials are available at https://seqml.github.io/simple.

## 1 INTRODUCTION

Distribution shift is a common problem in real-world applications, which breaks the independent and identically distributional (i.i.d.) assumption of machine learning algorithms (Wang et al., 2022). Mismatches between training and test distributions, which are quite common in reality, can largely deteriorate model performance and make machine learning models infeasible for practical applications (González & Abu-Mostafa, 2015). Therefore, enhancing the generalization ability of models has attracted increasing attention (Cha et al., 2021; Zhang et al., 2022).

For its practical significance, various methods have been proposed, e.g., domain alignment (Ganin et al., 2016; Gong et al., 2019; Arjovsky et al., 2019), meta-learning (Finn et al., 2017; Dou et al., 2019; Du et al., 2020), and ensemble learning (Mancini et al., 2018; Cha et al., 2021; Arpit et al., 2021). The effectiveness of DG algorithms is generally verified by fine-tuning a pre-trained ResNet(He et al., 2016) model with these algorithms (Gulrajani & Lopez-Paz, 2020). It has demonstrated that these algorithms improve upon empirical risk minimization (ERM) baseline on ResNet-50 backbone (Arpit et al., 2021; Wiles et al., 2021). Meanwhile, recent studies show that neural architectures and pretraining methods have a large impact on the model robustness to distribution shifts (Radford et al., 2021; Wiles et al., 2021). For example, vision transformers are more robust to texture and style shifts compared with ResNet-based models (Zhang et al., 2022), which are instead superior to transformer-based models on dense image classification tasks (Liu et al., 2022). In terms of pretraining, using pretraining datasets other than ImageNet-1k improves the generalization performance in one test domain, yet leads to performance degradation in another (Kim et al., 2022).

---

[*]It was conducted during the internship of Ziyue Li at Microsoft Research. Correspondence to Kan Ren.

These findings are in line with the No Free Lunch (NFL) Theorem (Wolpert, 1996), which suggests that no single model can always perform better than any other model without having substantive information about the targeted problem. In DG, we usually have very limited information about the test domain, so we are more likely to encounter the above challenge.

Inspired by these attempts, in this paper, we conduct a fine-grained study on the relationship between pretrained models and distribution shifts. From both empirical and theoretical evidence, we show that *no free lunch in terms of pretraining for domain generalization*, i.e., there is no single best pretrained model across shifting test distributions. Specifically, 283 pretrained models with different network architectures, pretraining datasets, and learning objectives are compared for their generalization performance under different distributional shifts. The results reveal that the pretrained models without fine-tuning generalize well to some unseen domains, but none of these models dominate in all unseen distributions. Furthermore, the theoretical analysis indicates that OOD generalization error is determined by the *fitness* between model (varying w.r.t. the network architectures and model weights) and test distribution. For any network architecture with fixed training distributions, such as pretrained models (Iandola et al., 2014; He et al., 2016; 2021a), it is always possible to find a beneficial or detrimental test distribution with a small or large generalization error.

Motivated by these findings, we propose an alternative DG paradigm that leverages pretrained models with different network architectures and shifting training distributions, upon which we match the most suitable pretrained models for each OOD sample. As shown in Figure 3, our paradigm (**s**pec**i**alized **m**odel-sam**ple** matching for domain generalization, SIMPLE) first adopts a simple label adapter that projects the label space of the pretraining domain[1] to that of unseen domains[2], where the adapter is shared by pretrained models from the same pretraining domain. Then, a matching network, which is aware of model specialty, selects a set of proper pretrained models and aggregates them together to conduct the prediction for each OOD sample. Notably, this promising alternative exhibits significant performance improvements, averaging 3.9% over the existing SOTA results, with gains of up to 12.2% on single datasets, and a significant increase in training efficiency.

To summarize, this work has made the following contributions:

• We theoretically and empirically analyze the generalization of pretrained models on shifting unseen test distribution, revealing no free lunch hypothesis exists that motivates our solution of model-sample matching.

• In complementary to traditional DG solutions, we propose a novel DG paradigm which directly leverages pretrained models without fine-tuning, and it has significantly improved the DG performance in the mainstream benchmark upon other strong baselines.

• Besides the performance gain, our method is even more efficient since it does not follow the common fine-tuning approach, shedding new light on using pretrained models in DG tasks.

## 2    No Free Lunch in Pretraining for Domain Generalization

In this section, we investigate if there exists free lunch in pretraining for DG, that is, whether we can find one single best pretrained model that generalizes across all distributional shifts. To this end, we first conduct an empirical analysis on the generalization ability of pretrained models over shifting distributions in Section 2.1, followed by a theoretical analysis in Section 2.2.

### 2.1    Generalizability analysis of the pretrained models

We here analyze the generalization ability possessed by different pretrained models. Note that existing DG methods generally adopt a specific ImageNet-pretrained model (e.g, ResNet-50), which has been shown not sufficient for generalization (Kumar et al., 2021; Kim et al., 2022). Thus, for a comprehensive analysis, we first incorporate 283 diverse pretrained models composed of diverse combinations of network architectures, pretraining datasets, objectives, and algorithms. Detailed information on all these models and more experimental settings are in Appendix A.4. For the efficient adaptation of pretrained models from pretraining domains to unseen domains, we propose to

---

[1]The data distribution where the pretrained models have been learned.
[2]Source and target domains in DG share the same label space yet differing from that of pretraining domains.

train only a label space adapter upon the pretrained models without modifying any pretrained model parameters, which is remarkably lightweight and will be elaborated detailedly in Section 3.2.

**Takeaway 1: Pretrained models possess decent generalization ability for some OOD samples.** By grafting such a lightweight label space adapter, we find that the generalization performance of the pretrained model is already promising in some cases. As a concrete example, given a fixed DenseNet-121 model (Iandola et al., 2014) pretrained on ImageNet, we employ an adapter that converts its prediction of the original ImageNet label space to that of OfficeHome (a dataset of DG benchmark) (Venkateswara et al., 2017), with the adapter training on source domains. This combination achieves an average accuracy of 78.3% on target domains, which is 5.9% higher than SOTA as detailed in Section 4.3.

**Takeaway 2: No dominant pretrained models across unseen domains.** Though pretrained models have possessed some generalization ability in some cases, however, their generalization performance relates to the specific unseen distributions. The left part of Figure 1 shows the relative performance of the pretrained models evaluated in different domains, with label adapter. Each domain represents a different data distribution. As can be seen, pretrained models vary greatly in performance on different test domains, without any single model being dominant in all domains.

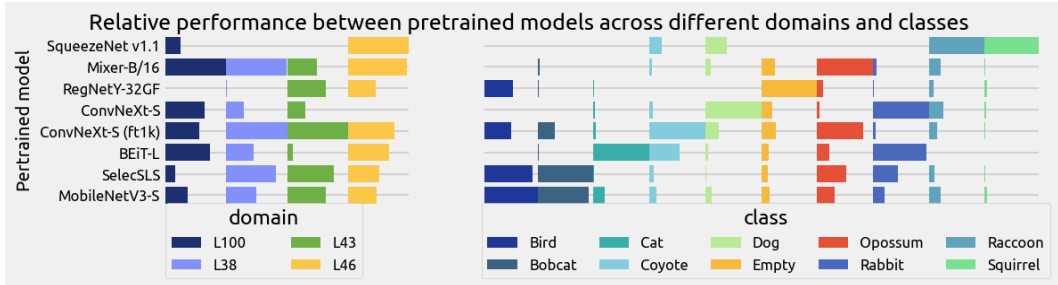

Figure 1: Classification performance comparison of pretrained models in different domains and different classes of the TerraIncognita dataset (Beery et al., 2018). For clarity of presentation, only partial results are shown. The complete results can be found in Appendix A.4.

**Takeaway 3: Pretrained models exhibit more diverse performance at finer-grained levels.** We further examine whether performance divergence also exists at a finer-grained level, such as class-level. The right part of Figure 1 presents the relative model performance on 10 classes in TerraIncognita dataset. Similar to that at domain-level, varying model performance among different classes is also noticeable. Moreover, there exists a more pronounced divergence in model performance at the finer class level, as evidenced in detail in Appendix A.4. It supports the necessity of incorporating pretrained models and being aware of their specialty at a fine-grained level, which motivates us to investigate and exploit the matching metric between models and test samples.

## 2.2 THEORETICAL ANALYSIS OF NO FREE LUNCH FOR DG

This section provides theoretical analysis to support the findings in Section 2.1. To alleviate the generalization difficulties associated with the train-test mismatch, mainstream DG methods seek domain-invariant features that can be generalized beyond the training distribution. However, there are several issues to consider. First, domain-invariant features learned from source domains can still be biased towards source domains and thus have limited performance on unseen domains (Cha et al., 2022). Second, domain-specific information has also been found to enhance DG, as it may be closely related to the sample labels that help generalize in certain unseen domains (Bui et al., 2021). These observations suggest that existing DG methods may not guarantee whether a model can generalize across distinct unseen target domains, the reason lies in the limitation of our knowledge of unseen domains, which is in line with NFL theorem. The analysis of the OOD generalization of kernel methods also points out the relevant insight that *shift in test distribution may help or hurt generalization* (Canatar et al., 2021).

**Theorem 1.** *(Informal; OOD Kernel Generalization;Canatar et al. (2021)) Given the kernel matrix $K(x, x')$ with Mercer decomposition $K(x, x') = \mathbf{\Phi}(x)^T \mathbf{\Lambda} \mathbf{\Phi}(x')$. Suppose the training data is i.i.d. generated from the distribution $p(x)$, the target function is given by $y = \bar{\mathbf{a}}^T \mathbf{\Phi}(x)$, and the training loss is kernel regression with ERM. The generalization error on distribution $\tilde{p}(x)$ is given*

by $E_g = E_g^{matched} + \kappa \bar{\mathbf{a}}^T (P\mathbf{\Lambda} + \kappa\mathbf{I})^{-1} \mathscr{O}' (P\mathbf{\Lambda} + \kappa\mathbf{I})^{-1} \bar{\mathbf{a}}$, where $\mathscr{O}_{ij} = \int dx \tilde{p}(x) \phi_i(x) \phi_j(x)$ and $\mathscr{O}' = \mathscr{O} - \frac{1-\gamma'}{1-\gamma} \mathbf{I}$. Here $E_g^{matched}$ is the generalization error when the training distribution and test distribution are matched (i.e., in-distribution error), $P$ is the number of training samples, $\kappa, \gamma$ are constants depend on $\mathbf{\Lambda}$, and $\gamma'$ is a constant depending on $\mathbf{\Lambda}$ and $\mathscr{O}$.

Theorem 1 will be detailed in Appendix A.5.

**Remark 1.** *(Interpretations of Theorem 1) Theorem 1 shows that $E_g$ is in the form $E_g = E_g^{matched} + \mathbf{v}\mathscr{O}'\mathbf{v}$, where the second term is due to distribution shift. Note that, if a pretrained model is kept frozen to serve as a static feature extractor for the subsequent trainable linear classifier, the pretrained model acts as a kernel and thus Theorem 1 applies.*

**Remark 2.** *(Generalization depends on the fitness between model and test distributions) The eigenvalue of matrix $\mathscr{O}$ depends on the alignment between the test distribution $\tilde{p}(x)$ and the kernel basis $\mathbf{\Phi}$ which is determined by the model (network architecture and model weights). The matrix $\mathscr{O}'$ may have negative eigenvalues when $\tilde{p}(x)$ and $\mathbf{\Phi}$ are matched well and OOD generalization is even better than i.i.d. generalization under such circumstances. On the contrary, when $\mathbf{\Phi}(x)$ is fixed, $\tilde{p}(x)$ can be adversarially chosen such that the matrix $\mathscr{O}'$ has large positive eigenvalues and thus the network fails to generalization to OOD data.*

**Remark 3.** *(No free lunch for a single model in DG) A major focus of DG is tackling covariate shift, where $\tilde{p}(x)$ can be set arbitrarily as long as $p(y|x) = \tilde{p}(y|x)$. Under covariate shift, no single pretrained model will outperform other models for all $\tilde{p}(x)$, implying no free lunch theorem in DG.*

The theoretical analysis above raises a natural question: *should not the focus be more on **matching** the pretrained model and testing distributions based on their fitness?* Given that pretrained models with fixed network architectures and training distributions always face beneficial or detrimental test distributions, we suggest incorporating more pretrained models to create diverse network architectures and shifting training distributions, to facilitate generalization. Depending on the fitness of the pretrained models to the target distributions, it is then possible to match OOD samples to appropriate models, thus bypassing the issue of using a single model indicated by the NFL theorem.

# 3 THE PROPOSED FRAMEWORK FOR DOMAIN GENERALIZATION

In this section, we present a novel framework, namely **s**pe**ci**alized **m**odel-sam**ple** matching for domain generalization (SIMPLE), that reformulates DG as a matching problem, in light of the analysis in Section 2. We first introduce the overall framework in Section 3.1. Then, we elaborate on the design of specialty-aware model-sample matching and ensemble in Section 3.2, followed by the learning algorithm in Section 3.3.

## 3.1 THE OVERALL FRAMEWORK

This section provides an overview of the SIMPLE framework, as shown in Figure 3. Following our analysis in Section 2, one single pretrained model is not sufficient to accommodate diverse OOD samples. From an intuitive view, it appears that *more* models need to be incorporated and *selected* to solve the problem. Analogous to recommending items (models) to users (samples) from the vast item set in recommender system, we formulate DG as a model-sample matching problem, with a model pool containing various models and a model dispatcher responsible for assigning these models to OOD samples appropriately.

**Preliminaries.** DG aims to tackle the shift of data distribution among different domains by transferring knowledge from seen to unseen domains. Unlike domain adaptation, samples from unseen target domain(s) are inaccessible in DG. For a domain, its input and label space can be denoted as $\mathcal{X} \in \mathbb{R}^d$ and $\mathcal{Y} \in \mathbb{R}^C$, respectively, where $d$ is the dimension of input and $C$ is the number of classes in $\mathcal{Y}$; And its samples are observed constructing a dataset $D = \{(x_i, y_i)\}_{i=1}^N$ with sample number $N$. Consider that we have $S$ source domains $\mathcal{D}_s = \{D_1, \ldots, D_S\}$ and $T$ target domains $\mathcal{D}_t = \{D_1, \ldots, D_T\}$ that share the same label space but with different joint distributions on $\mathcal{X} \times \mathcal{Y}$.

**Pretrained model pool.** As discussed in Section 2.2, the OOD generalization error depends on the fitness between the pretrained model and the test distribution, which is unknown in DG. Therefore, building a pool with diverse pretrained models is crucial for DG. Note that with abundant pretrained

models being released, it is easy to construct a model pool by simply downloading pretrained models from public repositories (Wightman, 2019), without further efforts such as retraining. We collect extensive pretrained models to serve for SIMPLE, as detailed in Appendix A.3. We denote a model pool with $K$ models as $\{f_k\}_{k=1}^K$, each of which is parameterized as $\theta_k$.

**Label space adapter.** As the label space of the pretraining domains generally differs from the one shared by source and target domains, label space adapters are required to make them consistent. The adapter is a linear mapping between these two label spaces (pretraining $\rightarrow$ source/target), which is shared among different pretrained models from the same pretraining domain (e.g., we only use one adapter for all the models trained on ImageNet-1k). Specifically, given pretrained model $f_k$, we parameterize the adapted model $h_\psi(f_k(\cdot; \theta_k))$ as $\theta'_k = [\psi; \theta_k]$, where $\psi$ denotes the parameters of the adapter, $h_\psi \in \mathcal{A}$: $\mathbb{R}^{C_o} \rightarrow \mathbb{R}^C$ and $C_o$ is the dimension of the label space of the original pretraining dataset of $f_k$. Through the label adapter, the output of pretrained model $f_k$ can be transformed and adapted to target domains as $\hat{y}_{ik} = h_\psi(f_k(x_i))$, without fine-tuning $\{\theta_k\}_{k=1}^K$. In this way, it can largely reduce the adaptation cost of the pretrained models onto the new domains.

By the above construction, SIMPLE differs from the existing methods in two aspects. First, most existing DG methods strive to fine-tune a specific pretrained model, which cannot generalize well to a variety of unseen domains, as analyzed in Section 2. Second, conventional options utilizing the pretrained models are fine-tuning and linear probing, as shown in Figure 2(A) and (B), respectively. In detail, fine-tuning the pretrained model is costly and can hurt generalization ability (Wortsman et al., 2022), while linear probing (i.e., replacing the last layer of the pretrained model and retraining that) may achieve better accuracy in OOD scenarios than fine-tuning the whole model (Kumar et al., 2021). However, the linear probing layer is not transferable across models, making the way either costly when a large number of models need to adapt to new target domains. Our label space adaption shown in Figure 2(C), instead, is remarkably light since it is shared by all the models from the same pretraining domain, which has also been empirically verified in Section 4.2.

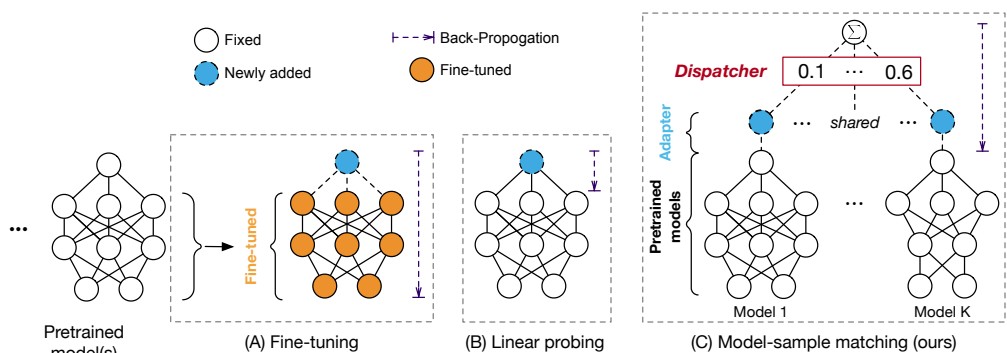

Figure 2: Different training paradigms in DG.

**Model dispatcher.** As shown in Section 2, there is no individual model performing the best across different tasks, thus, appropriate models need to be dispatched to address each specific task. We define a model dispatcher $g_\rho$, with parameter $\rho$, that takes the sample $x_i$ as input and determines the weight $w_k$ assigned to model $f_k$ for the sample $x_i$ with $\sum_{k=1}^K w_k = 1$. Here $w_k$ represents an estimate of the relative match between the model $f_k$ in the model pool and the sample.

Based on the constructed model pool and the model dispatcher, the prediction for each test sample is an ensemble of the predictions by the dispatched models. The final prediction is given by $\hat{y}_i = \sum_{k=1}^K [w_k \cdot h_\psi(f_k(x_i))]$. Finally, we can define a population loss as $\mathcal{E}_\mathcal{D}(\psi, \rho) = \frac{1}{|\mathcal{D}|} \sum_{j=1}^{|\mathcal{D}|} \mathbb{E}_{x_i \sim D_j} [l(\hat{y}_i, y_i)]$ over the given domain $\mathcal{D}$. The objective is to minimize the task-specific loss $l$ (e.g., cross-entropy loss for classification) over both source domains $\mathcal{D}_s$ and target domains $\mathcal{D}_t$ by only minimizing the empirical risk $\hat{\mathcal{E}}_{\mathcal{D}_s}(\psi, \rho)$ w.r.t. $\psi$ and $\rho$. The performance on the target domains, then, measures the generalization ability.

### 3.2 SPECIALTY-AWARE MODEL-SAMPLE MATCHING AND ENSEMBLE

Following the paradigm introduced in Section 3.1, this section elaborates on the model dispatcher, which consists of a model-sample matching network and a specialty-aware ensemble layer.

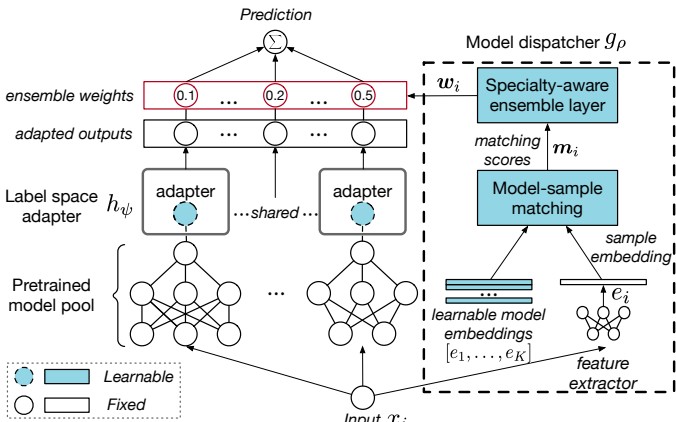

Figure 3: The SIMPLE framework. Based on a pool of fixed pretrained models, a recommender learns the matching of models and samples for model dispatching with the help of a label space adapter for prediction transformation. Note that only pretrained models from the same pretraining domain share the same label space adapter.

**Model-sample matching.** To capture the fitness of pretrained models towards OOD samples, a network is proposed to learn the model-sample matching function. We employ a simple recommendation algorithm (but not restricted), neural collaborative filtering (NCF) (He et al., 2017), to initiate a proof-of-concept for our idea. Following NCF, the matching scores $\boldsymbol{m}_i = [m_{i1}, \ldots, m_{iK}] \in \mathbb{R}^K$ of a sample $x_i$ and models $\{f_k\}_{k=1}^K$ are computed on their latent features $c_i$ and $\boldsymbol{c} = [c_1, \ldots, c_K]$, through a non-linear function activated multi-layer perceptrons (MLP) as $\boldsymbol{m}_i = \mathrm{MLP}(c_i^\top \boldsymbol{c})$. Specifically, the sample and the model are first embedded and then transformed into a joint latent space. The feature extractor of one pretrained model $f_{k_0}$ from our model pool, fixed as the sample feature extractor, generates sample embedding $e_i$ for the sample $x_i$. For the embedding of each model $f_k$, we introduce a learnable embedding $e_k$, included in $\rho$, which is randomly initialized and optimized during training. Both of them are processed by two non-linear functions activated MLP as $c_i = \mathrm{MLP}(\mathrm{MLP}(e_i))$ and $c_k = \mathrm{MLP}(\mathrm{MLP}(e_k))$, to map into the same space for matching scoring.

**Specialty-aware ensemble.** After constructing a series of ranked models based on the matching metric, the subsequent goal is to conduct prediction by selecting the most proper models. We argue that individual model prediction may not cover most of the target distribution, thus, instead of utilizing only the top-1 matched pretrained model, we propose to apply a specialty-aware model ensemble to derive the final prediction for the specific sample. It is also considered that ensemble has shown improved robustness (Lakshminarayanan et al., 2016) and model specialty in various tasks also plays a key role in prediction (Gontijo-Lopes et al., 2021), which will be further elaborated in Section 3.3. Specifically, we normalize matching scores by softmax function $\mathrm{Softmax}(z)_j = \frac{e^{z_j}}{\sum_{k=1}^K e^{z_k}}$ to highlight the relative competition among the pretrained models. That is, the ensemble weights $\boldsymbol{w}_i = [w_1, \ldots, w_k, \ldots, w_K] \in \mathbb{R}^K$, is computed as $\boldsymbol{w}_i = \mathrm{Softmax}(\boldsymbol{m}_i)$.

To save the inference time, we further select models with the top-$\tilde{k}$ ($\tilde{k} < K$) matching scores, before their inference for the given sample. In this paper, we set $\tilde{k} = 6$ and the sensitivity analysis on $\tilde{k}$ will be shown in Section 4.5. The overall inference cost is small as illustrated in Section 4.2.

### 3.3 OBJECTIVE AND LEARNING FOR SPECIALIZED MODEL-SAMPLE MATCHING

**Loss for ensemble learning.** The classification loss $\mathcal{L}_{\mathrm{ens}}(\psi, \rho) = l(\hat{y}_i, y_i)$ is optimized for the likelihood of final ensemble output, to update both the matching network and the adapter.

**Loss for label space adapter learning.** To train the general label space adapter $h_\psi$ for all pretrained models, we incorporate the weighted classification losses of adapted predictions of pretrained models to update the shared adapter, which is defined as follows:

$$\mathcal{L}_{\mathrm{adapter}}(\psi) = \sum_{k=1}^K w_k \cdot l(\hat{y}_{ik}, y_i) = \sum_{k=1}^K w_k \cdot l(h_\psi(f_k(x_i)), y_i). \tag{1}$$

**Loss for model specialty learning.** The model dispatcher generates ensemble weights to aggregate multiple model predictions for each sample, where models vary in their performance significantly over samples as indicated in Section 2. Thus, we expect to assign more weights to the models with

Table 1: DomainBed benchmarking. Baseline results are from original papers with the same setup.

| Algorithm | PACS | VLCS | OfficeHome | TerraIncognita | DomainNet | avg. |
|---|---|---|---|---|---|---|
| *Model Pool-A* | | | | | | |
| Non-ensemble algorithms | | | | | | |
| DANN (Ganin et al., 2016) | $84.6_{\pm1.1}$ | $78.7_{\pm0.3}$ | $65.4_{\pm0.6}$ | $48.4_{\pm0.5}$ | $38.4_{\pm0.0}$ | 63.1 |
| CORAL (Sun & Saenko, 2016) | $86.0_{\pm0.2}$ | $77.7_{\pm0.5}$ | $68.6_{\pm0.4}$ | $46.4_{\pm0.8}$ | $41.8_{\pm0.2}$ | 64.1 |
| MLDG (Li et al., 2018a) | $84.8_{\pm0.6}$ | $77.1_{\pm0.4}$ | $68.2_{\pm0.1}$ | $46.1_{\pm0.8}$ | $41.8_{\pm0.4}$ | 63.6 |
| MMD (Li et al., 2018b) | $85.0_{\pm0.2}$ | $76.7_{\pm0.9}$ | $67.7_{\pm0.1}$ | $49.3_{\pm1.4}$ | $39.4_{\pm0.8}$ | 63.6 |
| C-DANN (Li et al., 2018c) | $82.8_{\pm1.5}$ | $78.3_{\pm0.6}$ | $65.6_{\pm0.5}$ | $47.6_{\pm0.8}$ | $38.9_{\pm0.1}$ | 62.6 |
| ERM (Gulrajani & Lopez-Paz, 2020) | $85.7_{\pm0.5}$ | $77.4_{\pm0.3}$ | $67.5_{\pm0.5}$ | $47.2_{\pm0.4}$ | $41.2_{\pm0.2}$ | 63.8 |
| Fish (Shi et al., 2021) | $85.5_{\pm0.3}$ | $77.8_{\pm0.3}$ | $68.6_{\pm0.4}$ | $45.1_{\pm1.3}$ | $42.7_{\pm0.2}$ | 63.9 |
| LP-FT (Kumar et al., 2022) | $84.6_{\pm0.8}$ | $76.7_{\pm1.5}$ | $65.0_{\pm0.2}$ | $47.1_{\pm0.7}$ | $43.0_{\pm0.1}$ | 63.3 |
| MIRO (Cha et al., 2022) | $85.4_{\pm0.4}$ | $79.0_{\pm0.0}$ | $70.5_{\pm0.4}$ | $50.4_{\pm1.1}$ | $44.3_{\pm0.2}$ | 65.9 |
| Ensemble algorithms | | | | | | |
| SWAD (Cha et al., 2021) | $88.1_{\pm0.1}$ | $79.1_{\pm0.1}$ | $70.6_{\pm0.2}$ | $50.0_{\pm0.3}$ | $46.5_{\pm0.1}$ | 66.9 |
| EoA (Arpit et al., 2021) | **88.6** | 79.1 | 72.5 | 52.3 | 47.4 | 68.0 |
| MIRO + SWAD (Cha et al., 2022) | $88.4_{\pm0.1}$ | $79.6_{\pm0.2}$ | $72.4_{\pm0.1}$ | $52.9_{\pm0.2}$ | $47.0_{\pm0.0}$ | 68.1 |
| SIMPLE | $\mathbf{88.6}_{\pm0.4}$ | $\mathbf{79.9}_{\pm0.5}$ | $\mathbf{84.6}_{\pm0.5}$ | $\mathbf{57.6}_{\pm0.8}$ | $\mathbf{49.2}_{\pm1.1}$ | **72.0** |
| *Model Pool-B* | | | | | | |
| EoA$^+$ (Arpit et al., 2021) | 95.8 | 81.1 | 83.9 | 61.1 | 60.9 | 76.6 |
| MIRO + SWAD (Cha et al., 2022) | $96.8_{\pm0.2}$ | $81.7_{\pm0.1}$ | $83.3_{\pm0.1}$ | $\mathbf{64.3}_{\pm0.3}$ | $60.7_{\pm0.0}$ | 77.3 |
| SIMPLE$^+$ | $\mathbf{99.0}_{\pm0.1}$ | $\mathbf{82.7}_{\pm0.4}$ | $\mathbf{87.7}_{\pm0.4}$ | $59.0_{\pm0.6}$ | $\mathbf{61.9}_{\pm0.5}$ | **78.1** |

Table 2: Compare the training and inference cost of SIMPLE against that of SOTA baseline. ↑ (↓) means higher (lower) is better. The training time represents the overall back-propagation time. Details are in Appendix A.10.

| Pretrained model type | Algorithm | # Learnable params (↓) | Training time speedup v.s. SOTA baseline (↑) | Inference GFLOPs (↓) |
|---|---|---|---|---|
| ImageNet-pretrained | MIRO + SWAD (ResNet-50) | 25.6M | 1 × | 3.9 |
| | SIMPLE (Model Pool-A) | 0.9M | 1000 × | 9.6 |
| Diverse | MIRO + SWAD (RegNetY-16GF) | 79.7M | 1 × | 15.2 |
| | SIMPLE (Model Pool-B) | 0.9M | 1000 × | 9.6 |

higher sample-level specialties to achieve the best utilization of the pretrained models. We use the likelihood of the ground truth label $p(y_i \,|\, x_i; \theta'_k)$ on the $i$-th sample produced by model $f_k$ as the evaluation metric of its sample-level model *specialty*. That is, we try to minimize the estimation risk of the estimated model specialty on the ground truth, i.e., $w_k$ and $p(y_i \,|\, x_i; \theta'_k)$, as

$$\mathcal{L}_{\text{specialty}}(\rho) = -\sum_{k=1}^{K} \left[ p\left(y_i \,|\, x_i;\, \theta'_k\right) \cdot \ln(w_k) + (1 - p\left(y_i \,|\, x_i;\, \theta'_k\right)) \cdot \ln(1 - w_k) \right]. \quad (2)$$

$\mathcal{L}_{\text{specialty}}$ is used to optimize the model-sample network and the ensemble layer to work jointly as a specialty-aware model dispatcher.

Therefore, the total loss to minimize is $\mathcal{L} = a_e \mathcal{L}_{\text{ens}}(\psi, \rho) + a_d \mathcal{L}_{\text{adapter}}(\psi) + a_s \mathcal{L}_{\text{specialty}}(\rho)$, where $a_e$, $a_d$, and $a_s$ are loss weights. It is worth noting that the only parameters to update are $\{\psi, \rho\}$, each of which is lightweight compared to the pretrained models which remain fixed in our method, yet have been fine-tuned in other previous works.

## 4 EXPERIMENTS

### 4.1 EVALUATION PROTOCOL

We conduct experiments on DomainBed suite (Gulrajani & Lopez-Paz, 2020) which provides like-for-like comparisons between algorithms with a standard evaluation, as detailed in Appendix A.6.

***Datasets.*** We experiment on 5 real-world benchmark datasets including PACS (4 domains, 9,991 samples, 7 classes) (Li et al., 2017), VLCS (4 domains, 10,729 samples, 5 classes) (Fang et al., 2013), OfficeHome (4 domains, 15,588 samples, 65 classes) (Venkateswara et al., 2017), TerraIncognita (4 domains, 24,778 samples, 10 classes) (Beery et al., 2018), and DomainNet (6 domains, 586,575 samples, 345 classes) (Peng et al., 2019).

***Baselines.*** We compare SIMPLE with strong DG baselines including state-of-the-art. General DG methods incorporate elaborate learning algorithms including ERM (Vapnik, 1998), CORAL (Sun & Saenko, 2016), MLDG (Li et al., 2018a), MMD (Li et al., 2018b), DANN (Ganin et al., 2016), C-DANN (Li et al., 2018c), Fish (Shi et al., 2021), LP-FT (Kumar et al., 2021), and MIRO (Cha

Table 3: Results of (1) ablation study and (2) SIMPLE with different sized model pools.

| Algorithm | PACS | VLCS | OfficeHome | TerraIncognita | DomainNet | avg. |
|---|---|---|---|---|---|---|
| *Model Pool-A* | | | | | | |
| test-best single model | 62.1 | 66.6 | 78.3 | 40.5 | 28.3 | 55.2 |
| random ensemble | 40.2 | 48.6 | 34.7 | 16.2 | 12.0 | 30.3 |
| SIMPLE (small pool) | $84.1_{\pm0.5}$ | $79.8_{\pm0.1}$ | $79.9_{\pm0.1}$ | $56.8_{\pm0.2}$ | $46.3_{\pm0.4}$ | 69.4 |
| SIMPLE | $\mathbf{88.6}_{\pm0.4}$ | $\mathbf{79.9}_{\pm0.5}$ | $\mathbf{84.6}_{\pm0.5}$ | $\mathbf{57.6}_{\pm0.8}$ | $\mathbf{49.2}_{\pm1.1}$ | **72.0** |
| *Model Pool-B* | | | | | | |
| test-best single model | 95.6 | 82.1 | 78.3 | 40.5 | 52.7 | 69.8 |
| random ensemble | 38.5 | 46.5 | 28.2 | 19.7 | 12.5 | 29.1 |
| SIMPLE$^+$ (small pool) | $96.1_{\pm0.0}$ | $82.2_{\pm0.0}$ | $80.7_{\pm0.2}$ | $56.8_{\pm0.3}$ | $54.7_{\pm0.1}$ | 74.1 |
| SIMPLE$^+$ | $\mathbf{99.0}_{\pm0.1}$ | $\mathbf{82.7}_{\pm0.4}$ | $\mathbf{87.7}_{\pm0.4}$ | $\mathbf{59.0}_{\pm0.6}$ | $\mathbf{61.9}_{\pm0.5}$ | **78.1** |

et al., 2022). Some other works incorporate ensemble learning, including SWAD (Cha et al., 2021) and EoA (Arpit et al., 2021). MIRO has combined SWAD (MIRO + SWAD) into their approach, with the result being the current SOTA for DomainBed. More details can refer to Appendix A.7.

***Model pool composition.*** We collect 283 pretrained models to compose model pools, described in detail in Appendix A.3. Based on their pretraining domains, we divide them into a pure ImageNet-pretrained model pool (Model Pool-A) and one with pretrained models from different datasets (Model Pool-B). We denote SIMPLE using Model Pool-B as SIMPLE$^+$ to distinguish it from the one using Model Pool-A.

## 4.2 Evaluation results on DomainBed

This section presents the evaluation results of DomainBed suite, on which we compare SIMPLE with general DG algorithms to verify its effectiveness and efficiency. Specifically, we compare algorithms using ImageNet-pretrained models only (e.g., SIMPLE using Model Pool-A) and using pretrained models from diverse pretraining datasets (e.g., SIMPLE$^+$ using Model Pool-B), resp.

**Performance comparison.** As shown in Table 1, SIMPLE achieves an average performance of 72.0% with ImageNet-pretrained models, exceeding the current SOTA competitor (MIRO + SWAD) by 3.9% and ranks first in all datasets. SIMPLE even outperforms the second-best method by 12.2% on OfficeHome dataset. The results show that our proposed paradigm of using only pretrained models without fine-tuning is more effective than the traditional paradigm that requires fine-tuning. Furthermore, by using Model Pool-B that includes models pretrained on other datasets, SIMPLE$^+$ further improves the average performance by 6.1% over SIMPLE and beats other strong baselines.

**Training cost comparison.** As SIMPLE does not fine-tune these models but use the matching network to dispatch models, the training cost is almost negligible versus general DG methods. As shown in Table 2, the number of learnable parameters of SIMPLE (0.9M) is minor compared with the normal image backbone network used by MIRO (25.6M for ResNet-50 and 79.7M for RegNetY-16GF). In addition, training of SIMPLE uses noticeably less time with 1000 times speedup.

**Inference cost comparison.** Additionally, the inference cost is controlled through selectively activating appropriate models with the highest $\tilde{k}(< K)$ matching metric values, for inference. This reduces the inference cost to a large extent, though more than 200 pretrained models have been incorporated in the model pool. As shown in Table 2, although SIMPLE uses more inference cost than the model using ResNet-50 as the backbone, SIMPLE$^+$ uses less inference time than the existing SOTA (MIRO + SWAD) using RegNetY-16GF (Singh et al., 2022) as the backbone. That is, SIMPLE$^+$ *obtains new SOTA results with significantly higher training and inference efficiency*.

The significant training time advantage of SIMPLE and its surpassing performance indicate that SIMPLE is an effective and efficient paradigm for DG. We illustrate the matching preference of SIMPLE, i.e., which models are typically dispatched in specific domains, in Appendix A.11.

## 4.3 Ablation studies

We verify the effectiveness of SIMPLE design by answering two research questions (RQ) as below.

**(RQ1) Does SIMPLE achieve better generalization performance than the best single model in the model pool?** To verify, we train an individual adapter on source domains for each pretrained model and show the "cheating" upper bound of single model performance by reporting the *best single model performance on test set* as *test-best single model* in Table 3. As shown in Table 1, the best single model has outperformed the current SOTA (MIRO + SWAD) on certain datasets (e.g., OfficeHome and VLCS). Nevertheless, SIMPLE beats it in all the comparisons (**RQ1**). This implies

that SIMPLE has somehow compensated for the drawbacks brought by NFL of a single model in DG, by utilizing different pretrained models and dispatching them selectively to OOD samples.

**(RQ2) Is it necessary to assemble pretrained models according to their specialty?** The analysis in Section 2.1 shows that pretrained models possess certain generalization ability. Thus, a natural approach is to simply aggregate pretrained models without considering their relative specialty on different samples, e.g., randomly select $\tilde{k}$ models for each sample and average their outputs as the final prediction (Lakshminarayanan et al., 2017). The results of the random ensemble are shown in Table 3, illustrating that random ensemble lags behind SIMPLE by a large margin. This verifies the necessity to select and ensemble pretrained models based on their specialty over samples (**RQ2**).

### 4.4 PRACTICAL TIPS FOR CONSTRUCTING MODEL POOLS

In this section, we investigate the impact of different model pool properties on the generalization performance, to provide useful guidance for the composition and utilization of model pools. The corresponding results are presented in Table 3. Specifically, we focus on the size and diversity of model pools by comparing four different model pools, i.e., Model Pool-A-Small (ImageNet-pretrained, 15 models), Model Pool-A (ImageNet-pretrained, 244 models), Model Pool-B-Small (diverse pretraining datasets, 17 models), and Model Pool-B (diverse, 283 models). The composition of these model pools can be found in Appendix A.3.

**Tip 1: Use a larger model pool.** Based on the analysis in Section 2, a larger model pool is favored as it increases the probability that the model pool contains models that match well with each OOD sample. This is consistent with the comparison of Model Pool-A-Small with Model Pool-A, and Model Pool-B-Small with Model Pool-B. Both types of model pools show significantly better generalization performance as the size of the model pool increases.

**Tip 2: Incorporate diverse models.** The performance comparison of Model Pool-A(-Small) and Model Pool-B(-Small) indicates that adopting diverse pretrained models from different pretraining domains can enhance DG performance remarkably.

**Tip 3: Increasing model diversity matters more than increasing model pool size.** On top of Model Pool-A-Small, Model Pool-A incorporates 224 additional pretrained models from ImageNet pretraining domain, while Model Pool-B-Small adds 2 more models that are pretrained on YFCC100M (Radford et al., 2021). However, SIMPLE performs better upon Model Pool-B-Small, suggesting that the diversity of model pool may be more important for generalization.

The above tips from empirical observations that, more models and more diversity can improve generalization performance, again suggest that there is no free lunch for DG. Thus, different pretrained models are needed to address DG. SIMPLE provides a way to realize this goal both effectively and efficiently by (1) adapting pretrained models to unseen domains via label space adaption with a low cost and (2) dispatching best-fit models from a large model pool to handle each OOD sample.

### 4.5 SENSITIVITY ANALYSIS

We conduct sensitivity analysis for $\tilde{k}$ and sample feature extractor $f_{k_0}$, detailed in Appendix A.9, and the main findings include: (1) There is a marginal effect on the generalization performance improvement brought by increasing $\tilde{k}$, and SIMPLE can outperform SOTA baseline (MIRO + SWAD) even when $\tilde{k} = 2$; and (2) SIMPLE is robust to the selection of feature extractors.

## 5 CONCLUSION

Despite recent studies suggesting that network architectures and pretraining practices affect generalization ability to a large extent, no work has explored the use of these easy-to-obtain pretrained models to address domain generalization. Our work provides a comprehensive analysis of generalizing pretrained models to unseen domains and reveals that there is no free lunch of pretrained models in DG. Based on that, we propose a novel DG paradigm that leverages fixed pretrained models and dispatches them to OOD samples based on their matching metric to the target task. Extensive evaluations show that our proposal is a promising alternative for DG, with better generalization performance and significantly higher training efficiency compared to existing DG methods.

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

## A  APPENDIX

### A.1  RELATED WORK

Mainstream DG research generally addresses distributional shifts through data manipulation (Tobin et al., 2017; Peng et al., 2018), robust representational learning (Tzeng et al., 2014; Arjovsky et al., 2019), and exploration of new learning strategies such as ensemble learning Cha et al. (2021). We

here only review studies that are mainstream in DG and that relate to our method. For a more comprehensive survey, please refer to Wang et al. (2022).

**Data manipulation.** DG arises from the lack of sufficient data to feed machine learning models, so they are biased to the training distribution, resulting in an inability to work well on the test distribution (Wang et al., 2022). Thus, various methods resort to manipulating the training samples to simulate unseen target domains, among which there are two techniques, namely data augmentation (Shankar et al., 2018) and data generation (Li et al., 2021). Data augmentation has been widely used in general machine learning models to enhance their generalization ability by avoiding overfitting (Shorten & Khoshgoftaar, 2019). Inspired by its importance, Tobin et al. (2017) first uses it for DG to simulate test distributions, and subsequently, Peng et al. (2018); Khirodkar et al. (2019); Tremblay et al. (2018) also adopt data augmentation in various ways. In addition to the common feature augmentation, Peng et al. (2022) further propose to perform label augmentation. In the field of data generation, instead of augmentation, methods generate new samples or domains by means of such as Mixup (Zhang et al., 2017), auto-encoder (Qiao et al., 2020), generative adversarial network (Rahman et al., 2019).

**Representation learning.** In another promising direction, some DG studies attempt to extract domain-invariant features that can be generalized to unseen domains (Ben-David et al., 2006), or separate domain-shared and domain-specific parts from the features for generalization (Khosla et al., 2012). Motivated by the transferability of domain-invariant features across varied domains and the expectation of their generalization ability to unseen domains, methods try to obtain a specific feature space that is invariant to domain labels. Li et al. (2018b) first introduces adversarial training in DG, allowing the generator seeks to trick the discriminator about the domain labels of images and thus generate domain-invariant features. Several works (Shao et al., 2019; Rahman et al., 2020; Wang et al., 2020b) follow this direction. In the line of extracting domain-invariant features, other works explicitly align feature distribution learned from source domains, with the differences measured by Wasserstein distance (Zhou et al., 2020), maximum mean discrepancy (Wang et al., 2018; 2020a), or mean and variance (Peng et al., 2019). At a meta level, Balaji et al. (2018) consider learning a regularization function on the classifier to avoid biasing to domain-specific information. In contrast, for robust representation learning, some work aims to separate domain-invariant variables from the specific ones of the learned features (Niu et al., 2015; Ilse et al., 2020; He et al., 2021b).

**Ensemble learning.** Ensemble learning methods Hansen & Salamon (1990); Zhou et al. (2018); Li et al. (2023) exploit multiple models to produce prediction results and combine the results with various techniques, e.g., boosting Schapire (1990); Freund (1995); Moghimi et al. (2016) or mean aggregation Zhou et al. (2018); Zhang et al. (2020), aiming at achieving better performance than individual model alone. As a well-known technique to improve generalization performance, ensemble learning has also been explored in DG. One general idea of using ensemble learning by existing DG methods is to utilize the relationship between unseen domains and source domains. Specifically, in the work of Mancini et al. (2018), they train an individual classifier for each specific source domain and one additional classifier to predict the probability of each domain that a sample belongs to, and aggregate their predictions by weights accordingly. In addition, Segu et al. (2020) proposes to maintain domain-specific batch normalization layers, which will be weighted for aggregation in inference. Zhou et al. (2021) instead trains only domain-specific classifier heads while letting them share the same feature extractor. SWAD (Cha et al., 2021) and EoA (Arpit et al., 2021) instead use model ensemble and weight ensemble directly to improve the generalization. Though promising in enhancing performance, ensemble learning is criticized for its computational efficiency, which hinders the practical application (Wang et al., 2022).

*Relation to existing work.* The main difference between the existing DG works and SIMPLE resides in the fact that these efforts seek to find an optimal model that can settle all distributional shifts. Even DG algorithms that use ensemble learning are trying to find an optimal ensemble that has a flatter loss landscape and better generalization ability (Cha et al., 2021). To this end, existing works use a specific pretrained model for initialization, as this performs better than training from scratch (Wortsman et al., 2022), and then fine-tune it with training algorithms like data manipulation, robust representation learning, or ensemble learning as discussed above. In contrast, recent studies have shown that such specific pretrained models may not be sufficient to solve distributional shifts, and that pretraining strategies need to be chosen for different types of distributional shifts. To the best of our knowledge, SIMPLE is the first work to address DG by directly using different pretrained models without fine-tuning, by reformulating DG as a model-sample matching problem. Moreover,

SIMPLE does not fine-tune these pretrained models, but uses domain-level label space adaptation to achieve effective adaptation, which has significantly reduced the cost of training and adaptation.

## A.2 MORE DISCUSSION ABOUT SIMPLE

Towards a robustness algorithm that can effectively and efficiently leverage extensive pretrained models for DG, there are certain challenges that remain in our paper. Currently, SIMPLE is not feasible for adding new pretrained models into the model pool for direct utilization. Although re-training is lightweight, we need a more straightforward approach to address this problem due to the rapid development of pre-trained models. We leave this problem, i.e., the cold-start problem which has been long studied in recommendation fields, as further work.

## A.3 PRETRAINED MODEL POOLS

This section presents the pretrained models used in SIMPLE. With more and more pretrained models being published, it is straightforward to build a pretrained model pool consisting of several public pretrained models for direct adaptation to novel domains. In particular, these models are categorized according to their pretraining domain as an ImageNet-pretrained model pool (Model Pool-A) and a pool (Model Pool-B) containing models pretrained on datasets other than the ImageNet dataset.

**Model Pool-A.** DG methods commonly use ImageNet-pretrained models to initialize the model weights for further fine-tuning(Kim et al., 2021). To present fair comparisons with those methods, we first construct an ImageNet-pretrained model pool (Model Pool-A). Specifically, we collect 244 models with diverse network architectures and learning objectives from popular 3rd-party repositories[3][4][5]. Note that, the models pretrained on the same dataset, i.e., ImageNet, will share the same label adapter transforming the vanilla model outputs to the target label space. Therefore, for Model Pool-A, only one label space adapter will be trained, to transform the original predictions for 1,000 classes in ImageNet into the predictions of the target class space.

**Model Pool-B.** Previous studies also found that ImageNet-pretrained model (e.g, ResNet-50) is not sufficient for generalization (Kumar et al., 2021; Kim et al., 2022), suggesting the need for models pretrained on different datasets. Therefore, we further expand the Model Pool-A by incorporating models that trained on datasets including ImageNet-21k, YFCC100M (Radford et al., 2021), Instagram 3.6B (Singh et al., 2022), as Model Pool-B. In total, we add 39 models, whose weights are obtained from the same sources or from their public repositories[6][7]. Thus, Model Pool-B contains 283 pretrained models with varied network architectures, learning objectives, and pretraining datasets.

To analyze the impact of model pool size on performance, we take a portion of the models from these two model pools and use them to construct Model Pool-A-Small and Model Pool-B-Small. Specifically, in **Model Pool-A-Small**, we incorporate the architectures including AlexNet (1) (Krizhevsky et al., 2012), DenseNet-121/169/201 (3) (Iandola et al., 2014), Dual-Path-Network-68 (1) (Chen et al., 2017), NASNetMobile (1) (Zoph et al., 2018), ResNet-18/34/50 (3) (He et al., 2016), SE-ResNet-50 (1) Hu et al. (2018), SqueezeNet-1.0/1.1 (2) (Iandola et al., 2016), and MAE-ViT-Base/Large/Huge (3) (He et al., 2021a). On the other hand, several DG algorithms also use models pretrained on other datasets, such as 1G-1B (Arpit et al., 2021) and ILSVRC12 (Thomas et al., 2021). Therefore, we build **Model Pool-B-Small** which contains two more CLIP models (Radford et al., 2021), ViT-B/16 and ViT-B/32, which are trained on a subset of the YFCC100M dataset of roughly the same size as ImageNet.

A detailed list of pretrained models is given in table 7, with the sources of the models, the dimensions of their outputs and FLOPs (floating point operations per second).

---

[3] https://github.com/rwightman/pytorch-image-models
[4] https://github.com/Cadene/pretrained-models.pytorch
[5] https://github.com/facebookresearch/mae
[6] https://github.com/openai/CLIP
[7] https://github.com/facebookresearch/SWAG

A.4 EMPIRICAL EVIDENCE OF NO FREE LUNCH

In this section, we elaborate on the empirical analysis in Section 2, and provide more experimental details and insights for the analysis.

**Transfer and performance measurement of pretrained models.** As discussed in Section 2, instead of fine-tuning or conducting linear probing to adapt those pretrained models to be a predictive model for unseen target domains, we only train a label space adapter that learns the mapping function $h_\psi$. Then we train this shared adapter $h_\psi$ with the empirical loss $\hat{\mathcal{E}}_{D_s}(\psi, \rho)$ without fine-tuning the pretrained model parameters $\{\theta_k\}$, as formally described in Section 3.2. With the trained adapter, we use the likelihood of the ground truth label $p(y_i \mid x_i; \theta_k')$ on the $i$-th sample produced by each adapted model, which also indicates the confidence of the ground truth label $y_i$ of the model and $\sum_{y \in \mathcal{Y}} p(y \mid x_i; \theta_k') = 1$. We utilize this likelihood as the evaluation metric of its sample-level model *specialty*.

**Experimental settings.** First, we analyze the specialty distribution of each pretrained model from an aggregation view (i.e, domains and classes, respectively), and we verify if there exists a dominant pretrained model that generalizes best across different unseen domains. We calculate domain-level model specialty as summation of the sample-level specialty over all domain samples as $\sum_{(x_i, y_i) \sim D} \log p(y_i \mid x_i; \theta_k')$, on TerraIncognita (Beery et al., 2018) with four domains. To reflect the relative model performance, we perform min-max normalization for model specialty values in the same domain. These complete results are shown in Figure 4, with only partial results put in Section 2 for a clearer presentation. Then, we further examine whether performance divergence also exists at the finer-grained class-level. Similar as that at domain-level, Figure 4 presents the relative model performance on 10 classes in TerraIncognita. In addition, to clearly compare model specialty differences at the two levels, we present heatmaps of specialty differences (measured by Kullback-Leibler divergence) for domain and class pairs, respectively in Figure 5.

Based on the results of comparing the performance of all pretrained models on different domains and classes, we can draw more empirical insights as follows:

(1) Most importantly, there is no single model that performs best both across domains and classes;

(2) The divergence of the performance distribution is significantly more prominent at the class level than at the domain level, as evidenced by the comparison of the Figure 5(a) and (b);

(3) As a side finding, models that perform well in the 'Bird' class generally perform well in the 'Bobcat' class, as indicated by their pair-wise Kullback-Leibler divergence values. However, models that perform well in 'Bird' and 'Bobcat' usually perform poorly in 'Dog' and 'Rabbit' classes. This can be viewed as another piece of evidence of NFL for shifting distribution shifts.

A.5 NO FREE LUNCH THEOREM

We here detail the formal version of Theorem 1 with related definitions.

Considering a kernel regression task, training samples $\{x^\mu, y^\mu\}$ are sampled from i.i.d. and the labels which contain noisy are generated from a target function $y^\mu = \bar{f}(x^\mu) + \epsilon^\mu$ where the noise covariance is $\langle \epsilon^\mu \epsilon^\nu \rangle = \varepsilon^2 \delta_{\mu\nu}$. Then the model for regression is trained by minimizing the empirical error (ERM loss):

$$f^* = \arg\min_{f \in \mathcal{H}} \frac{1}{2} \sum_{\mu=1}^{P} (f(x^\mu) - y^\mu)^2 + \Lambda \langle f, f \rangle_{\mathcal{H}},$$

where $\mathcal{H}$ is a Reproducing Kernel Hilbert Space (RKHS) associated with a positive semi-definite kernel $K(x, x') : \mathbb{R}^D \times \mathbb{R}^D \to \mathbb{R}$, and $\langle ., . \rangle_{\mathcal{H}}$ is the Hilbert inner product. The generalization error on the test distribution $\tilde{p}(x)$ is $E_g(\mathcal{D}) = \left\langle \left( f^*(x) - \bar{f}(x) \right)^2 \right\rangle_{\tilde{p}(x)}$, which is a random variable whose value depends on the sampled training dataset. Therefore, the generalization error is averaged over the distribution of all datasets with sample size $P$ :

$$E_g = \left\langle \left( f^*(x) - \bar{f}(x) \right)^2 \right\rangle_{\tilde{p}(x), \mathcal{D}}.$$

Based on the problem setting, Canatar et al. (2021) proposes the following proposition:

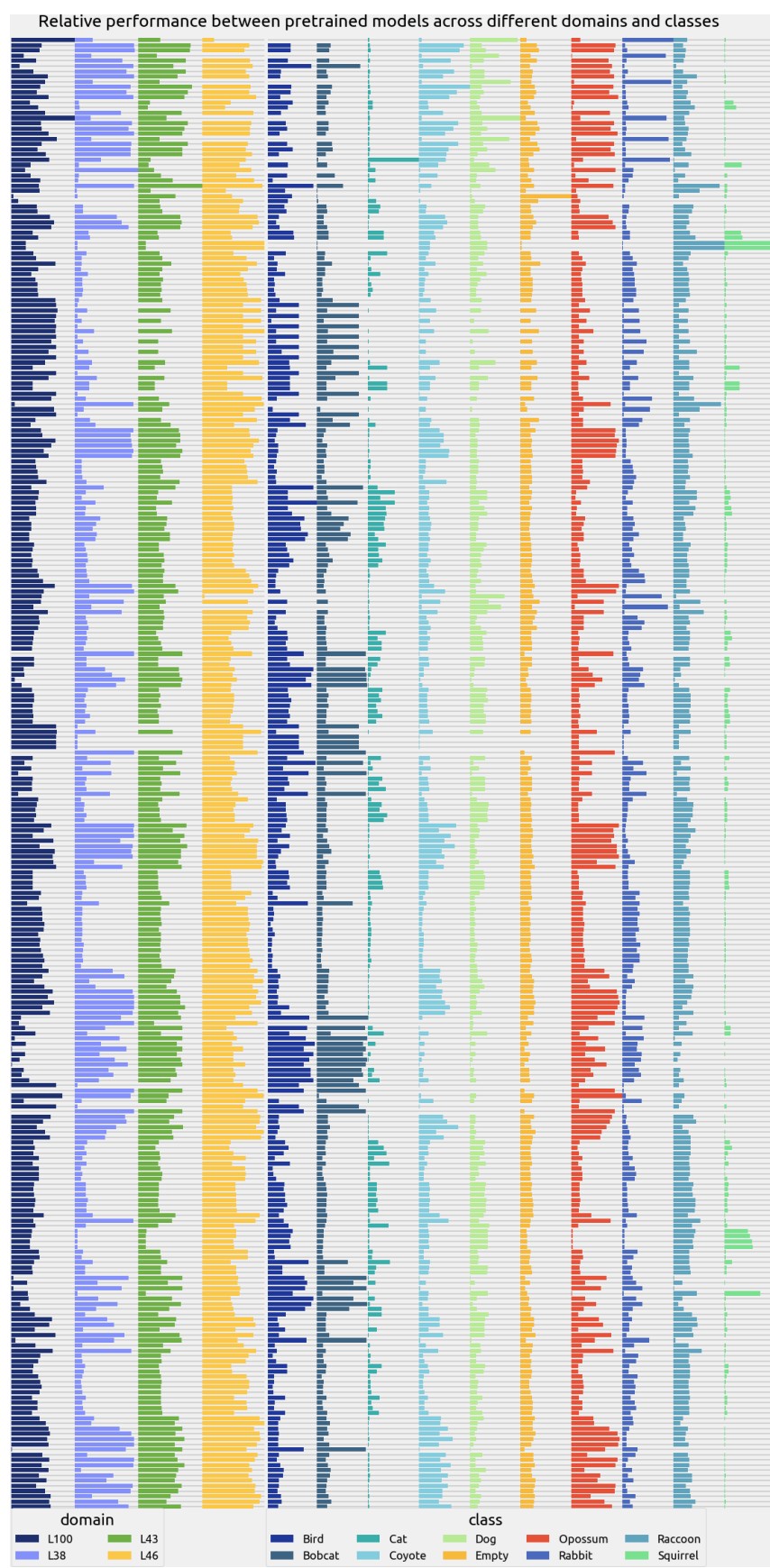

Figure 4: Performance comparison of all pretrained models on different domains and classes of the TerraIncognita benchmark. Note that we omit the model names for clarity, and each row represents the performance of one model.

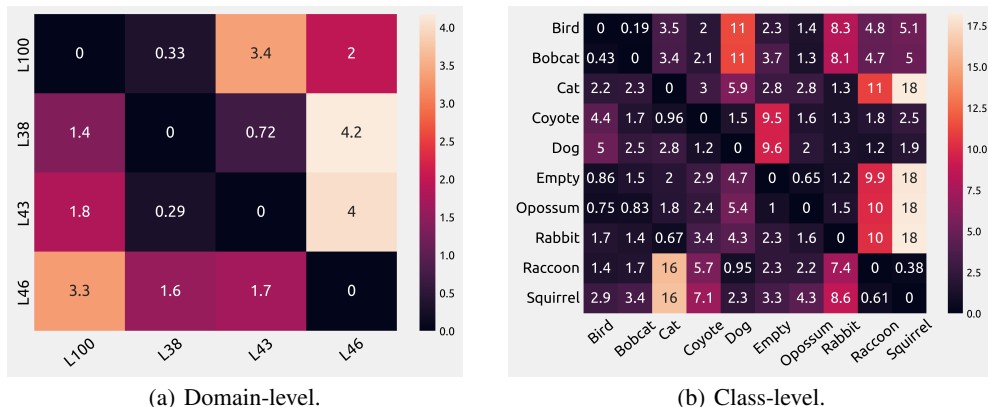

(a) Domain-level.                                        (b) Class-level.

Figure 5: Kullback-Leibler divergence between the performance distribution over all the leveraged pretrained models in domain-leval (a) and class-level (b), respectively.

**Proposition 1.** *(Restating Proposition 1 in Canatar et al. (2021)) Consider the kernel regression problem defined above, where the model is trained on $p(x)$ and tested on $\tilde{p}(x)$. Consider the Mercer decomposition of the RKHS kernel $K(x, x') = \mathbf{\Phi}(x)^T \mathbf{\Lambda} \mathbf{\Phi}(x')$, where we defined $M \times M$ (M possibly infinite) diagonal eigenvalue matrix $\mathbf{\Lambda}_{\rho\gamma} = \eta_\rho \delta_{\rho\gamma}$ and the column vector $\mathbf{\Phi}(x) = (\phi_1(x), \phi_2(x) \ldots, \phi_M(x))$, with $\int dx p(x) \mathbf{\Phi}(x) \mathbf{\Phi}(x)^\top = \mathbf{I}$. Also consider an expansion of the target function in the Mercer basis $f(x) = \bar{a}^\top \mathbf{\Phi}(x)$ with a coefficient vector $\bar{a}$. The dataset averaged OOD generalization error is given by:*

$$E_g = \underbrace{E_g^{0,p(x)}}_{\text{ID error}} + \underbrace{\frac{\gamma' - \gamma}{1 - \gamma}\varepsilon^2 + \kappa^2 \bar{a}^\top (P\Lambda + \kappa \mathbf{I})^{-1} \mathscr{O}'(P\Lambda + \kappa \mathbf{I})^{-1}\bar{a}}_{\text{distribution shift error}},$$

$$\kappa = \Lambda + \kappa \operatorname{Tr}\left(P + \kappa\Lambda^{-1}\right)^{-1}, \quad \gamma = P \operatorname{Tr}\left(P + \kappa\Lambda^{-1}\right)^{-2}, \quad \gamma' = P \operatorname{Tr}\mathscr{O}\left(P + \kappa\Lambda^{-1}\right)^{-2},$$

*where $\kappa$ must be solved self-consistently, and we defined the $M \times M$ overlap matrix*

$$\mathscr{O}_{\rho\gamma} = \int dx \tilde{p}(x)\phi_\rho(x)\phi_\gamma(x), \quad \mathscr{O}' = \mathscr{O} - \frac{1 - \gamma}{1 - \gamma}\mathbf{I}.$$

*Here $E_g^{0,p(x)}$ denotes the generalization error when both training and test distributions are matched to $p(x)$ (i.e., in-distribution error) and is given by:*

$$E_g^{0,p(x)} = \frac{\gamma}{1 - \gamma}\varepsilon^2 + \frac{\kappa^2}{1 - \gamma}\bar{a}^\top (P\Lambda + \kappa\mathbf{I})^{-2}\bar{a}.$$

*Further, the expected estimator is:*

$$\langle f^*(x; P)\rangle_D = \sum_\rho \frac{P\eta_\rho}{P\eta_\rho + \kappa}\bar{a}_\rho \phi_\rho(x).$$

A.6 EVALUATION SETTINGS

**Evaluation protocol.** We follow DomainBed protocol to conduct our evaluation, for fair comparisons with baselines. We use the *training-domain validation set* protocol for model selection. Specifically, one domain in a dataset is selected as the target domain and the rest as source domains, from which 20% of samples are used as the validation set. All runs are repeated 3 times using different random seeds, thus, with different train-validation splits. The out-of-domain test performance averaged over all domains will be reported for each dataset. In addition, we use the standard number of iterations of 5,000 for all datasets, with early-stop based on validated accuracy.

**Hyperparameter tuning.** Here we state the details of hyperparameter tuning in our experiments. We use separate Adam optimizers (Zhang, 2018) for the ensemble network and label space adapter. Table 4 lists the hyperparameters to tune and their search space. For each domain, we sweep through 48 different hyperparameter settings.

Table 4: Hyperparameters we set or tune for SIMPLE.

| Hyperparameter | Search Range |
|---|---|
| learning rate used for matching network | [0.0001,0.1] |
| learning rate used for label space adapter | [0.0001,0.1] |
| loss weight $a_e$ | [0.01, 1.0] |
| loss weight $a_d$ | [0.01, 1.0] |
| loss weight $a_s$ | [0.01, 1.0] |
| weight decay of Adam optimizer | [1e-3,1e-4,1e-5,1e-6, 1e-7] |
| batch size | [32, 64, 128, 256] |

## A.7 BASELINES

- Stochastic Weight Averaging Densely (SWAD) (Cha et al., 2021): SWAD performs weight ensemble during model training.

- Ensemble of Average (EoA) (Arpit et al., 2021): EoA combines both model ensemble and weight ensemble by taking an ensemble of moving average models from 6 runs. They experiment with three different pretrained models as initialization. The first one is pretrained on ImageNet with ResNet-50 and the second is pretrained on both ImageNet and a much larger additional dataset, IG-1B, with a more advanced backbone, ResNeXt-50 (Xie et al., 2017). Additionally, with a pretrained RegNetY-16GF (Singh et al., 2022), EoA achieves its best results. We denote the last one as EoA$^+$ to indicate it uses the additional dataset and to compare with SIMPLE$^+$.

- Random ensemble: In contrast to SIMPLE of learning to select models for ensemble, we also compare it with an average ensemble of $\tilde{k}$ models chosen randomly for each sample.

## A.8 MORE ABLATION STUDY

Here we introduce an additional baseline that uses the same set of ensemble weights for all samples in a domain, rather than generating different ensemble weights to aggregate model predictions differently for each OOD sample.

For ensemble-based approaches, the overall prediction is given by $\hat{y}_i = \boldsymbol{w}^\top [f_1(\boldsymbol{x}_i), \cdots, f_K(\boldsymbol{x}_i)]$, where $w_k$ is the weight for aggregating the prediction of $k$-th models $f_k(\boldsymbol{x}_i)$. In SIMPLE, the ensemble weights are given by $\boldsymbol{w} = \text{MLP}(\boldsymbol{c}_i^\top \boldsymbol{C})$, where $\boldsymbol{c}_i$ is the embedding for the $i$-th sample and $\boldsymbol{C}$ contains the embeddings for all the models.

Therefore, there is a special case of proposed ensemble approach, where the ensemble weights $\boldsymbol{w} \in \mathbb{R}^K$ are randomly initialized and optimized through back-propagation. Here in this simplified version, $\boldsymbol{w}$ is shared for all the samples in the dataset and does not incorporate model or sample information. Compared with this simplified version, SIMPLE explicitly leverages sample and model information (encoded in the embedding vectors $\boldsymbol{c}_i$ and $\boldsymbol{C}$) to generate specialized ensemble weights for each sample, which is more fine-grained. With model and sample embedding, SIMPLE enjoys much lower training cost when incorporating new pretrained models that are not in the model pool.

We implement this special case and conduct experiments on OfficeHome dataset to compare with SIMPLE. By performing sufficient hyperparameter tuning, such a simple version achieves an average accuracy of 82.3% on OfficeHome, which is worse than 87.7% of SIMPLE (and even worse than 84.6% of SIMPLE using the much smaller Model Pool-A) and 83.9% of the existing SOTA. It in contrast suggests that, incorporating information about models and the sample to conduct fine-grained model-sample matching in SIMPLE, is necessary and more effective, than using a set of weights and optimized by back-propagation.

## A.9 SENSITIVITY ANALYSIS

### A.9.1 ANALYSIS OF $\tilde{k}$ IN TOP-$\tilde{k}$ MODEL SELECTION

Intuitively, using more models in an ensemble may lead to better performance (Zhang et al., 2020). However, a larger ensemble size also means that the ensemble consumes more computation in infer-

ence. Thus, in inference, we choose to selectively activate $\tilde{k}$ ($< K$) models with higher ensemble weights. It is necessary to verify the impact of how many models are used for the prediction of each sample on the final performance, which can help balance the effectiveness and efficiency of our approach.

**Settings.** To demonstrate the sensitivity of changes in the parameter $\tilde{k}$ (the number of models activated in inference) to the final generalization performance, we evaluated the performance of SIMPLE$^+$ with different values of $\tilde{k}$ in the top-$\tilde{k}$ selection on the OfficeHome dataset. Specifically, we evaluate $\tilde{k} \in [1, \ldots, 10]$, with a limited and average sweep of the hyperparameters to save computational time. Therefore, the accuracy may not be optimal.

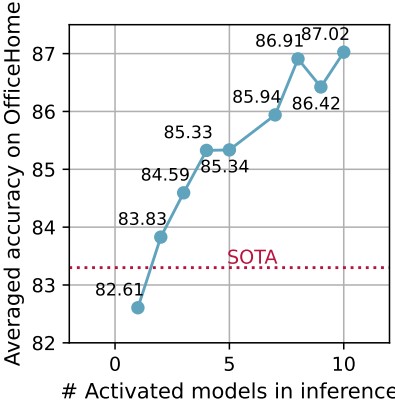

Figure 6: The impact of $\tilde{k}$ values of top-$\tilde{k}$ selection in inference, on the generalization performance of SIMPLE$^+$ on the OfficeHome dataset.

As illustrated in Figure 6, the highest accuracy of 87.02% is obtained when the number of activated models reaches the highest value we set. This indicates that the generalization performance can benefit from using more models to compose the ensemble for the final prediction. However, another important finding of this result is that the marginal effect of increasing the number of models decreases as the number of activated models increases. For example, when $\tilde{k}$ is increased from 1 to 2, SIMPLE$^+$ gains 1.22% performance gain. In contrast, when $\tilde{k}$ shifts from 8 to 10, the performance gain is only 0.11%. Therefore, it is predictable that since matching networks can provide a decent ranking of pretrained models, we can obtain promising generalization performance with limited activation models to save computational cost. Note that when $\tilde{k} = 2$, our method has obtained a performance that exceeds the existing SOTA (MIRO + SWAD).

### A.9.2 ANALYSIS OF FEATURE EXTRACTOR SELECTION

We perform an ablation study to verify the robustness of our matching network to the selection of feature extractors. Here we provide details for that claim in Section 4.5.

**Settings.** In general, ResNet-based models are used as feature extractors. We compare the performance of our method here with a ResNet-based feature extractor (ResNet-34) and a more advanced network structure (EfficientNet-B7 with Noisy Student) to see if these changes result in significant performance differences. Specifically, ResNet-34 and EfficientNet-B7-NS obtain ImageNet top-1 classification accuracy of 75.0% and 86.9%, respectively. This ablation study is performed on the OfficeHome dataset, with results shown in Table 5.

Table 5: Performance comparison of SIMPLE$^+$ using ResNet-34 and EfficientNet-B7-NS as the feature extractor.

| domain | A | C | P | R | avg. |
|---|---|---|---|---|---|
| ResNet-34 | $\mathbf{88.5}_{\pm0.2}$ | $76.7_{\pm0.6}$ | $\mathbf{92.8}_{\pm0.9}$ | $\mathbf{92.6}_{\pm0.3}$ | $\mathbf{87.7}_{\pm0.5}$ |
| EfficientNet-B7-NS | $85.8_{\pm0.8}$ | $\mathbf{77.0}_{\pm0.7}$ | $92.2_{\pm0.5}$ | $91.9_{\pm0.7}$ | $86.7_{\pm0.6}$ |

As can be seen, using ResNet-34 or EfficientNet-B7-NS results in similar performance, without one dominating in all domains. Therefore, feature extractor selection has little impact on the generalization performance of SIMPLE, showing its robustness. Meanwhile, it is noted that the number of parameters of EfficientNet-B7-NS is three times higher than that of ResNet-34. Therefore, we choose ResNet-34 because it can provide good performance with less computation cost.

### A.9.3    ANALYSIS OF LABEL SPACE ADAPTER TRAINING

As detailed in Section 3.3, the label space adapter of SIMPLE is trained by two losses, $\mathcal{L}_{\text{ens}}(\psi, \rho)$ and $\mathcal{L}_{\text{adapter}}$, and the training process will be influenced by the ensemble weights $\boldsymbol{w}_i$. In this section, we analyze and verify whether such an influence affects the training of adapters.

Note that, we train the same adapter for all the pretrained models from the same pretraining domain, which is more efficient (than training different adapters for different pretrained models) and also avoids unexpected training instability. When considering the large model pool with various models from different pretraining domains (e.g., Model Pool-B), we believe the influence is acceptable or even beneficial since: (1) adapters will observe all the training samples, so their training is sufficient; (2) since the number of parameters of adapters is relatively small, the training of adapters will not have a significant impact on the final performance; (3) although the training of adapters is influenced by the ensemble weights, it is unbiased because the ensemble weights are optimized according to the final objective.

Furthermore, we conduct experiments to verify whether training label space adapters under the influence of ensemble weights will affect the training of adapters or degrade the performance.

**Settings.** We compare single model performance with and without the influence of ensemble weights on the corresponding label space adapter. For *no influence on adapters*, we separately train individual adapters for each model. We select several models which are assigned with small ensemble weights in SIMPLE. The evaluation is on '*art*' domain of the OfficeHome dataset.

Figure 7 shows the performance difference between (1) a model with the adapter trained with and without ensemble weights (blue bars); and (2) our method SIMPLE and the SOTA baseline for reference (orange bar). As can be seen, (1) single model performance does not significantly degrade under the influence of ensemble weights (even less than the difference between our method SIMPLE and the existing SOTA); (2) moreover, for some models in Figure 7, training a shared adapter (i.e., influenced by the ensemble weights) can make a single model perform even better. The possible reason is that training a shared adapter for multiple models avoids overfitting, leading to better generalization.

### A.10    TRAINING AND INFERENCE COST COMPARISON

We compare the training and inference costs of SIMPLE with general DG methods, including SOTA and methods that use ensemble learning. Specifically, for training costs, we evaluate the number of learnable parameters and the overall training time. And for inference, we compute GFLOPs. To fairly compare training costs, we run experiments of ERM, SWAD, and SIMPLE on a single Nvidia Tesla V100 and compare their overall back-propagation time from the start of training to the end (or early-stop). And based on the statistics of ERM and SWAD, we estimate the training time for EoA and MIRO, respectively. The results are shown in Table 6.

**Training cost comparison.** As shown in Table 2, training SIMPLE paradigm uses noticeably less time. SIMPLE$^+$ takes only 0.1% of the time of ERM on PACS. The significant training time advantage of the method and its surpassing performance suggest that SIMPLE is an effective and efficient paradigm for domain generalization.

**Inference cost comparison.** Although ensemble methods like EoA and SEDGE achieve better generalization performance at the cost of higher inference FLOPs, SEDGE still manages to save a large amount of inference cost compared to the previous best ensemble model (half of the inference FLOPs compared to EoA). This is because SIMPLE only selects models with the highest $\tilde{k}(< K)$ ensemble weights. Therefore, only $\tilde{k}$ of $K$ models are activated for inference per sample, which reduces the inference cost to a large extent. In addition, compared to SOTA (MIRO + SWAD), which uses a larger network architecture to achieve better results than using ResNet-50, SIMPLE$^+$

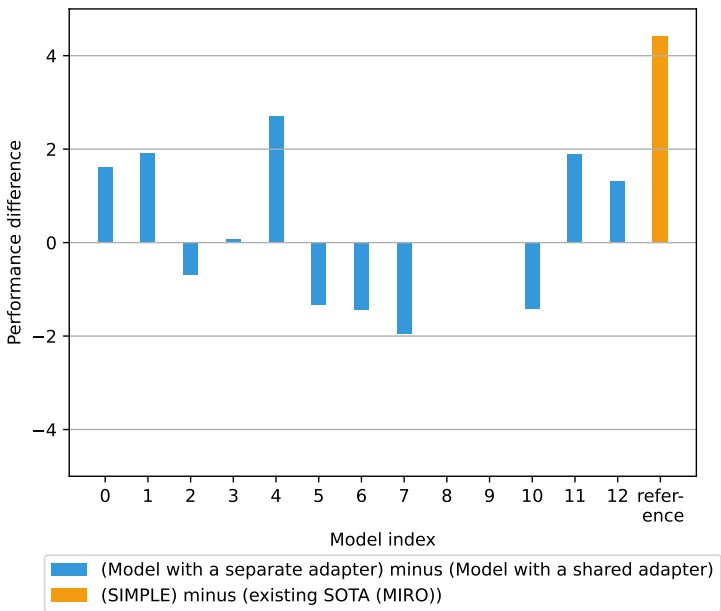

Figure 7: The difference of single model performance with and without the influence of ensemble weights, on the OfficeHome dataset.

keeps the inference cost using significantly lower GFLOPs than SOTA uses to obtain the new SOTA results.

Table 6: The comparison of training and inference cost. Here the training time implies the overall back-propagation time. The training times of ERM and SWAD are derived from the statistics of our runs, and we estimate the training time of EoA and MIRO based on that of ERM. The run for SWAD on DomainNet failed due to out-of-memory.

| Algorithm | # Learnable parameters (↓) | Training time (↓) | | | | | Inference GFLOPs (↓) |
| --- | --- | --- | --- | --- | --- | --- | --- |
| | | PACS | VLCS | OfficeHome | Terra | DomainNet | |
| ERM with ResNet-50 | 25.6M | 3.4h | 3.6h | 3.3h | 3.4h | 9.8h | 3.9 |
| SWAD with ResNet-50 | 25.6M | 2.1h | 2.4h | 2.1h | 2.1h | / | 3.9 |
| EoA with ResNet-50 | 153.4M | 20.4h | 21.6h | 19.8h | 20.4h | 58.8h | 23.5 |
| MIRO + SWAD with ResNet-50 | 25.6M | 3.4h | 3.6h | 3.3h | 3.4h | 9.8h | 3.9 |
| MIRO + SWAD (SOTA) with RegNetY-16GF | 79.7M | 10.6h | 11.2h | 10.3h | 10.6h | 30.5h | 15.2 |
| SIMPLE with Model Pool-A | 0.9M | 9.2s | 12.0s | 16.9s | 9.6s | 55.9s | 9.7 |
| SIMPLE$^+$ with Model Pool-B | 0.9M | 8.8s | 10.3s | 20.9s | 9.3s | 79.1s | 9.6 |

## A.11   MATCHING PREFERENCE ANALYSIS

SIMPLE shows excellent generalization performance in the case of dispatching only fixed pre-trained models to predict each OOD sample. Thus, we are curious about its matching preference, that is, which models are typically dispatched in specific domains. This could also provide insights into which types of pretrained models might be more suitable for certain domains.

**Settings.** We attempt to analyze the preferences of our approach for network architectures and pretraining datasets. To do so, we first measure the importance of the pretrained models and then

perform a refined analysis of their architectures and pretraining datasets. Specifically, by taking the model dispatcher trained by SIMPLE, we compute the sum of the ensemble weights assigned to each pretrained model as its importance measure for ranking. Then, we classify these pretrained models according to the basic architecture and the pretraining dataset. Treating the ranking of the pretrained models as the ranking of the associated types, we can measure the importance of each type by calculating the mean reciprocal rank (MRR).

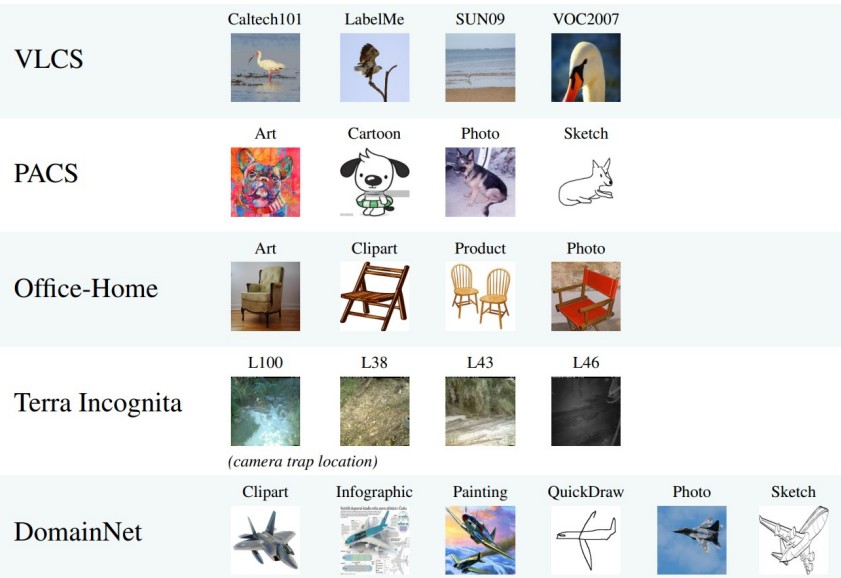

Figure 8: Samples that are included in DomainBed, from Table 3 in Gulrajani & Lopez-Paz (2020).

**Analysis of network architecture.** Figure 13, 14, 15, 16, and 17 show the raw rankings of the pretrained models on different datasets. From the ranking information, we can see that SIMPLE assigns markedly different pretrained models to samples from different domains. This may imply that these domains, as shown in Figure 17, may differ widely from each other and thus need to be handled by varying combinations of pretrained models. We then classify these pretrained models based on their basic architecture, i.e., CNN-based, ViT-based, and MLP-based, with their MRR values over different datasets and domains shown in Figure 9. It can be seen that ViT-based models are dispatched more frequently than CNN-based models in most domains, with the exception of 6 domains. Observations over these domains suggest that CNN-based models tend to be used more for real images (e.g., photo), while ViT-based models are chosen to handle stylistic or textural variations (e.g., sketch).

**Analysis of pretraining datasets.** We then analyze on the pretraining datasets, including ImageNet-1k, ImageNet-21k[8], CLIP (Radford et al., 2021), and SWAG (Singh et al., 2022), used for pretrained models. The MRR results of these different pretraining datasets on five datasets are shown in Figure 10, with scores grouped by domain. On one hand, it can be seen that our method shows a *preference for SWAG, CLIP, and ImageNet-21k over ImageNet-1k dataset*. This preference can be supported by recent studies finding that pretraining on ImageNet-1k is insufficient for generalization (Kumar et al., 2022) and that changing the dataset may improve generalization performance (Kim et al., 2022). On the other hand, despite this preference, SIMPLE dispatches different models for each domain (e.g. on TerraIncognita-L100 neither SWAG nor CLIP is chosen much, but rather ImageNet-pretrained models are used), suggesting that there is still *no free lunch for DG in the selection of pretraining datasets*.

---

[8]https://image-net.org/index

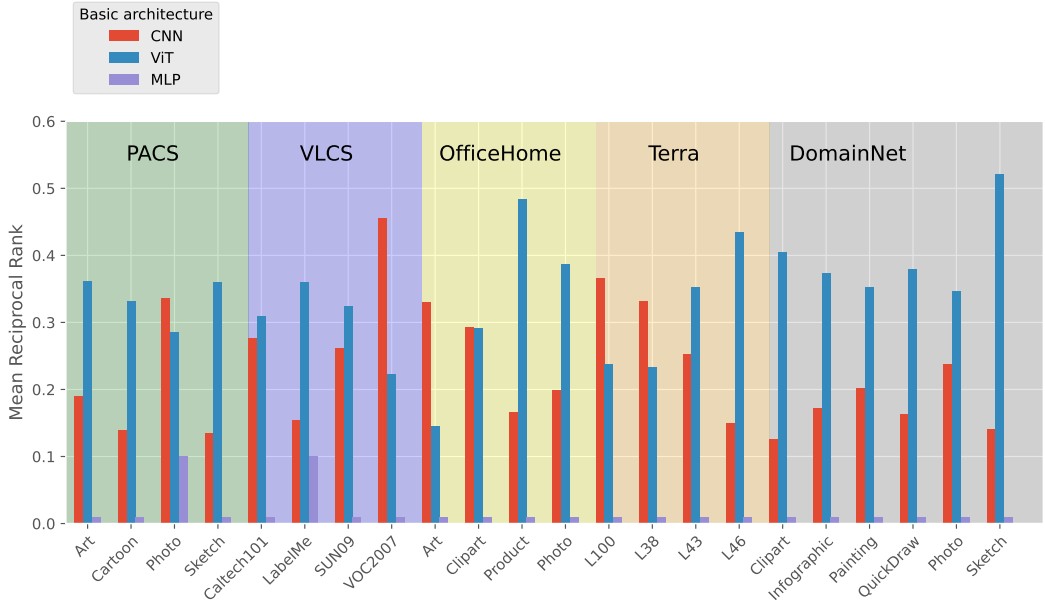

Figure 9: MRR values for the different network architectures used by the pretrained models.

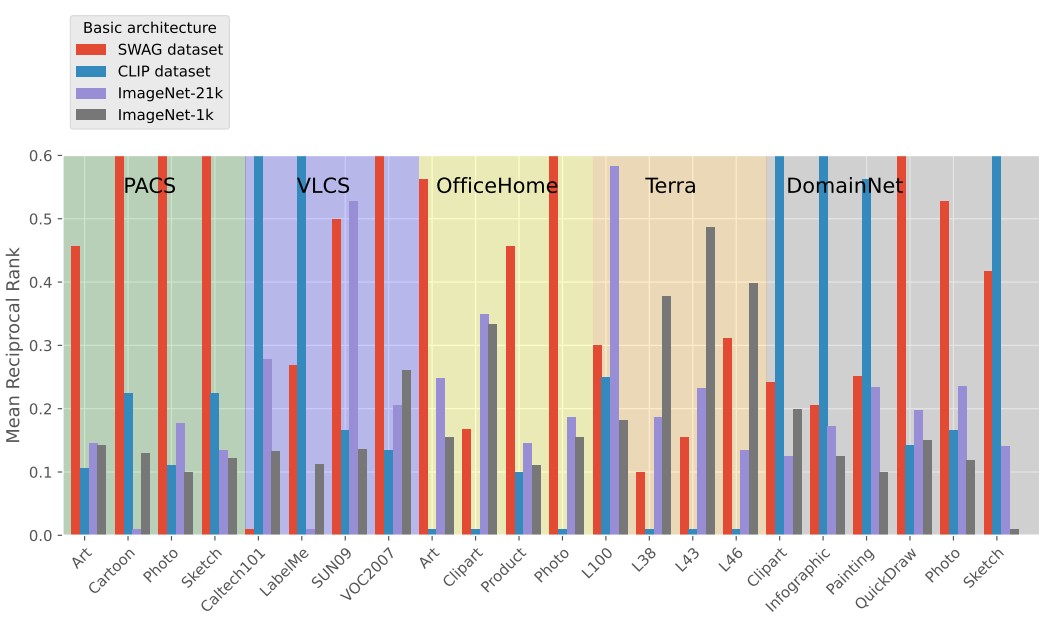

Figure 10: MRR values for the different pretraining datasets used by the pretrained models.

### A.12 LEVERAGING MATCHING PREFERENCE AS TRANSFERABILITY MEASUREMENT

In this section, we leverage the matching preferences of the learned model dispatcher to optimize the model pool size and study the performance of SIMPLE for different model pool sizes.

Specifically, since the model dispatcher of SIMPLE learns to match the best suitable models for unseen samples at a meta-level, its matching preference over unseen domains (without accessing ground truth labels) can be regarded as a measurement of model transferability on these unseen samples. Therefore, we can further optimize the model pool size used for each domain in a two-stage manner: (1) first learn with a large model pool on source domains to generate matching preferences; and (2) then reconstruct a smaller model pool to include models preferred by our dispatcher.

**Settings.** We experiment this two-manner training on PACS and OfficeHome datasets, using different reconstructed model pool sizes in the second stage. Note that we construct a model pool specific to each domain in a dataset, based on aggregated matching preferences for all samples of the unseen domain.

The results of PACS and OfficeHome datasets are shown in Figure 11 and 12, respectively. As can be seen, given that SIMPLE can automatically match models more suitable for transferring to unseen domains, we can actually go beyond the existing SOTA approach even using a model pool with only two models (while the best single model fails to outperform since there is No Free Lunch). And when using a smaller model pool, SIMPLE can even further improve the performance of the OfficeHome dataset.

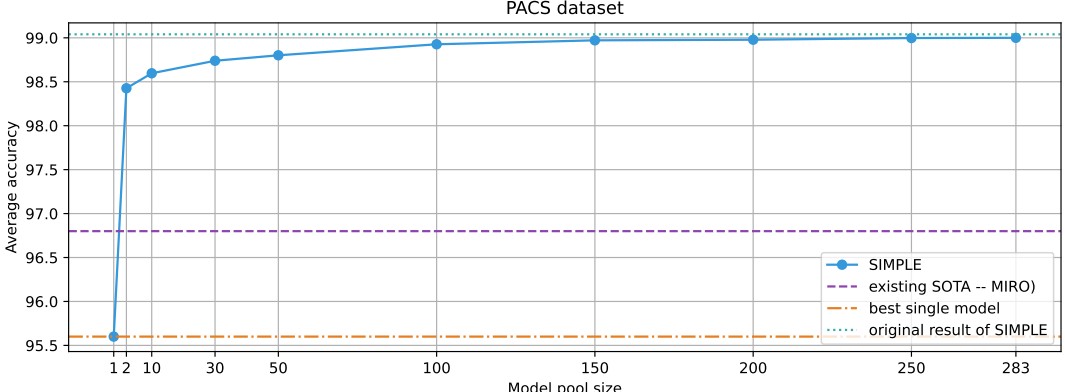

Figure 11: The performance of SIMPLE on PACS dataset, with model pools of different sizes.

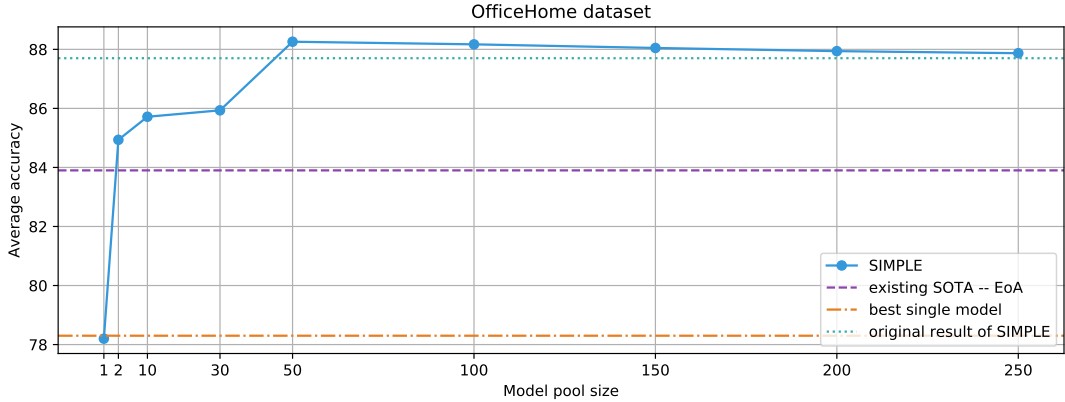

Figure 12: The performance of SIMPLE on OfficeHome dataset, with model pools of different sizes.

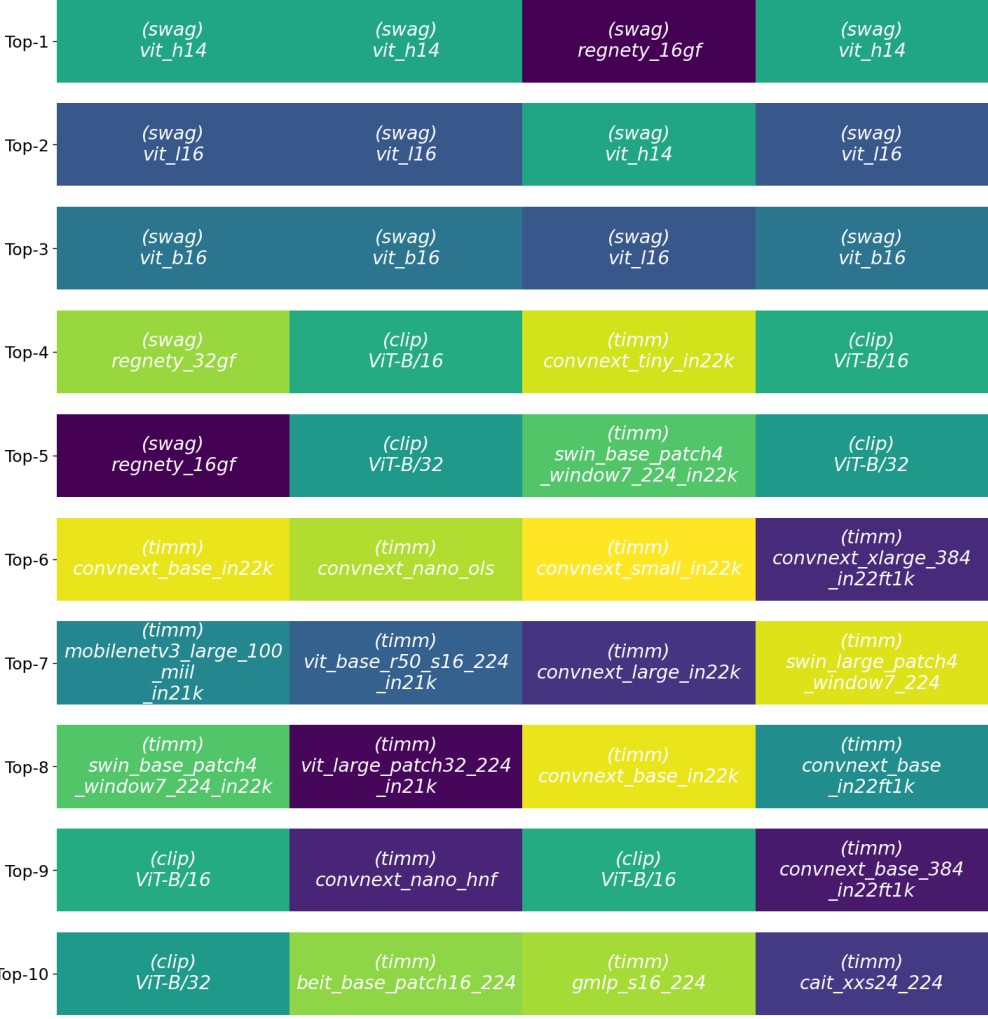

Figure 13: Ranking of the models in the four domains of PACS using the sum of the ensemble weights assigned to the models. The four columns from left to right in the figure correspond to the different domains (Art, Cartoon, Photo, Sketch) in the dataset.

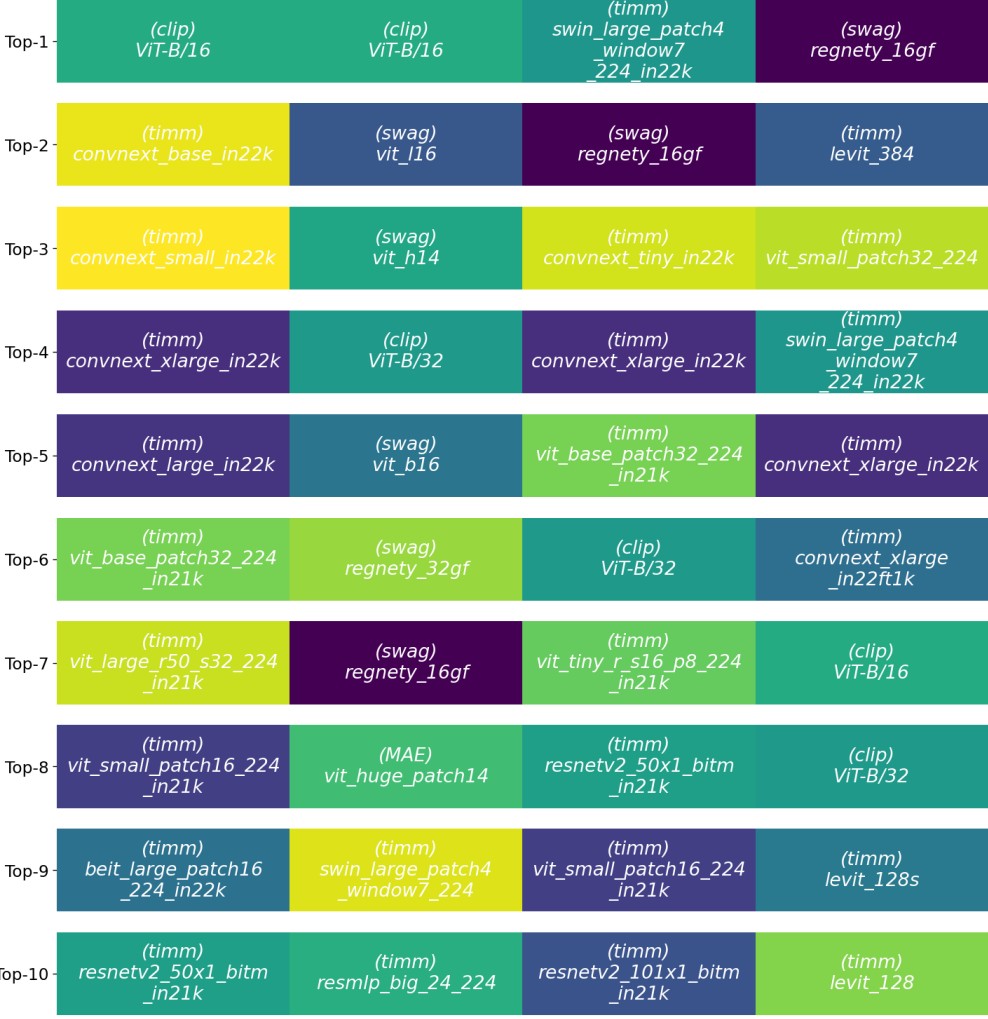

Figure 14: Ranking of the models in the four domains of VLCS using the sum of the ensemble weights assigned to the models. The four columns from left to right in the figure correspond to the different domains (Caltech101, LabelMe, SUN09, VOC2007) in the dataset.

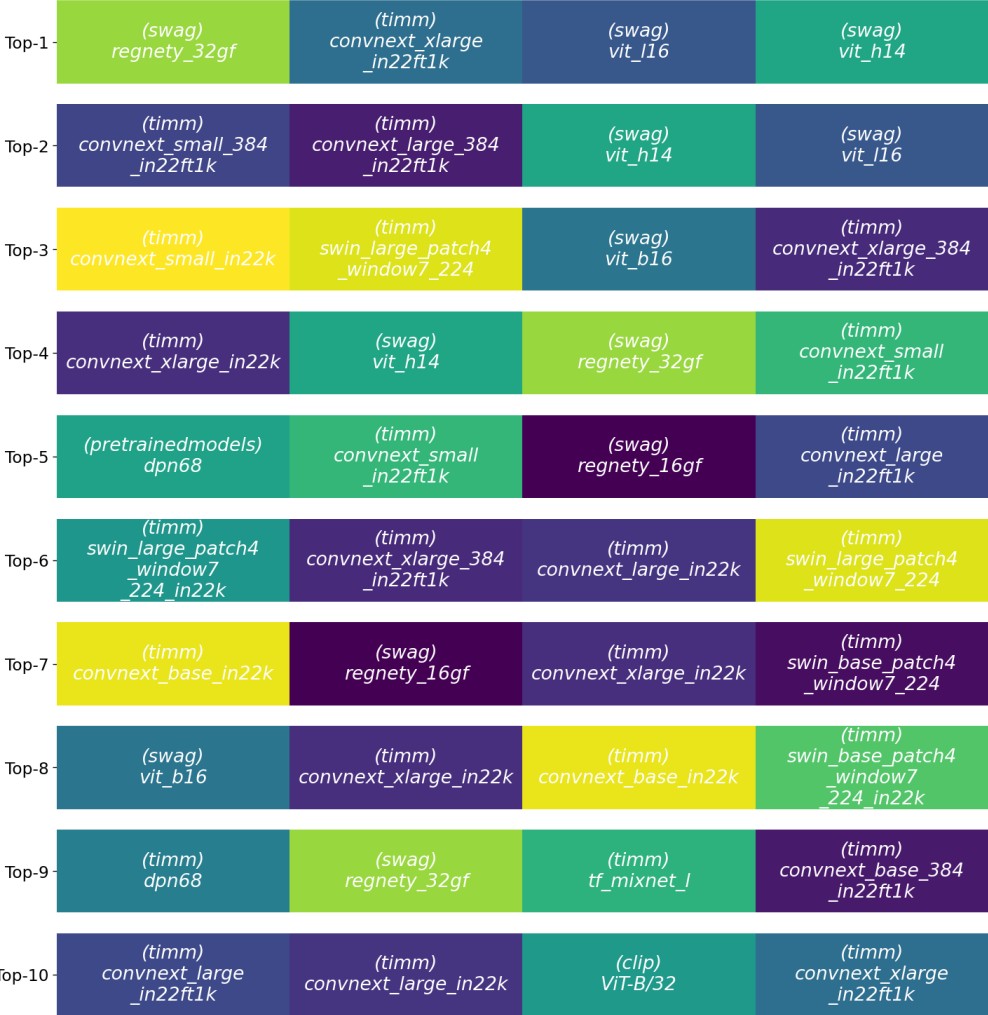

Figure 15: Ranking of the models in the four domains of OfficeHome using the sum of the ensemble weights assigned to the models. The four columns from left to right in the figure correspond to the different domains (Art, Clipart, Product, Photo) in the dataset.

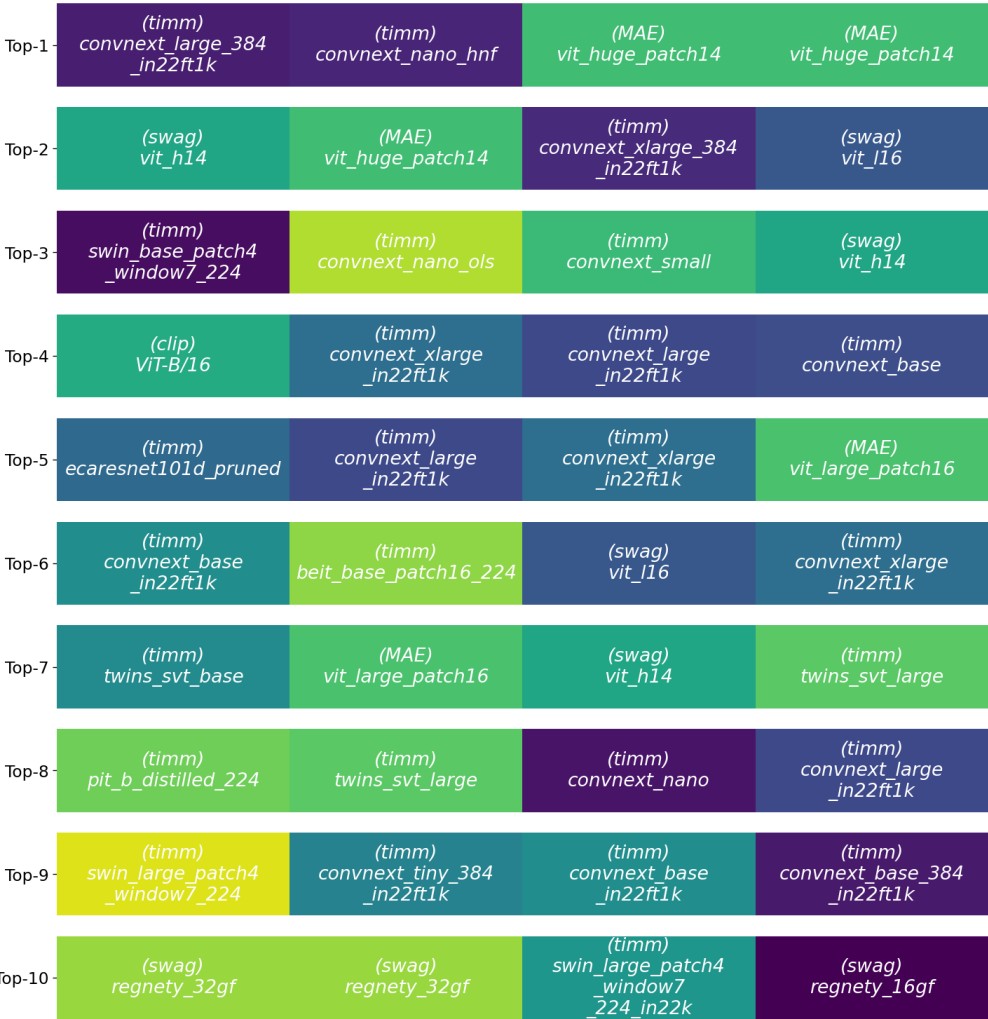

Figure 16: Ranking of the models in the four domains of TerraIncognita using the sum of the ensemble weights assigned to the models. The four columns from left to right in the figure correspond to the different domains (L100, L38, L43, L46) in the dataset.

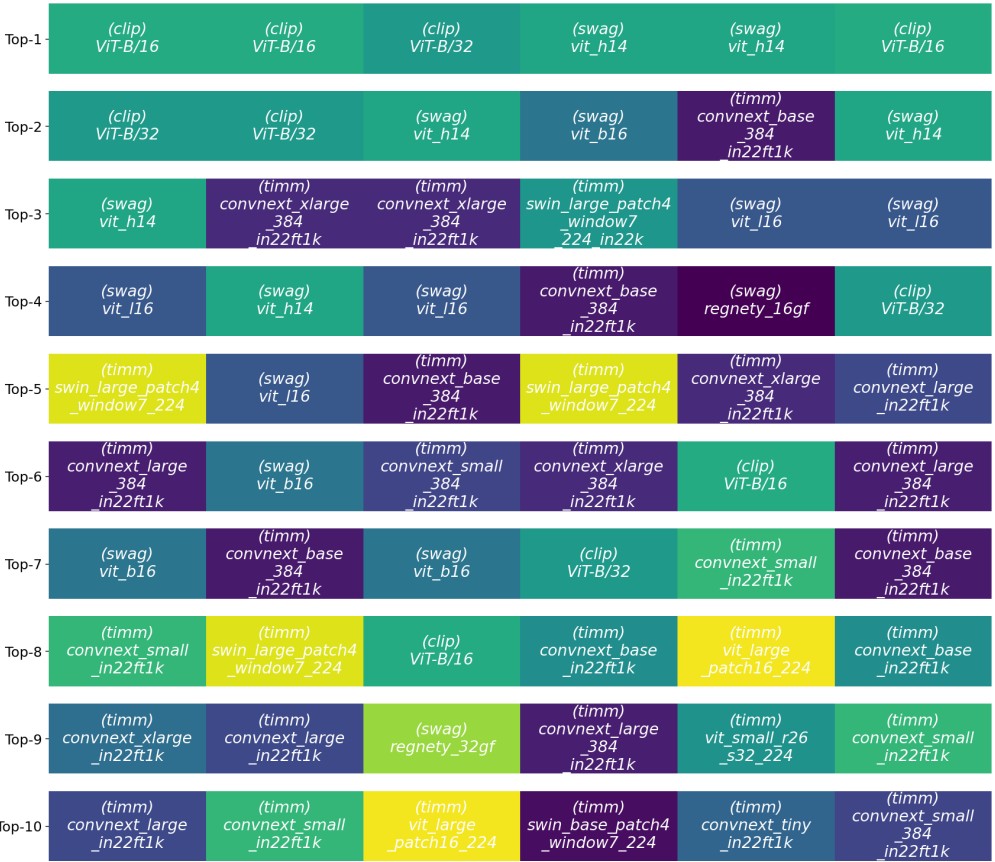

Figure 17: Ranking of the models in the four domains of DomainNet using the sum of the ensemble weights assigned to the models. The six columns from left to right in the figure correspond to the different domains (Clipart, Infographic, Painting, QuickDraw, Photo, Sketch) in the dataset.

Table 7: List of pretrained models from one of the following sources: (timm, pretrainedmodels, clip, MAE, or SWAG). The output dimension is the number of classes in the pretraining domain.

| Model id | Source | Model Name | Output Dimension | MFLOPs |
|---|---|---|---|---|
| 1 | pretrainedmodels | squeezenet1_0 | 1000 | 785.29 |
| 2 | pretrainedmodels | squeezenet1_1 | 1000 | 335.60 |
| 3 | timm | resnetv2_152x2_bit_teacher | 1000 | 4.10 |
| 4 | timm | resnext50_32x4d | 1000 | 4062.06 |
| 5 | timm | resnext50d_32x4d | 1000 | 4292.47 |
| 6 | timm | rexnet_100 | 1000 | 379.98 |
| 7 | timm | rexnet_130 | 1000 | 631.64 |
| 8 | timm | rexnet_150 | 1000 | 835.36 |
| 9 | timm | rexnet_200 | 1000 | 1458.16 |
| 10 | timm | selecsls42b | 1000 | 2843.06 |
| 11 | timm | selecsls60 | 1000 | 3425.65 |
| 12 | timm | selecsls60b | 1000 | 3461.41 |
| 13 | timm | semnasnet_100 | 1000 | 297.21 |
| 14 | timm | seresnet50 | 1000 | 3923.46 |
| 15 | timm | seresnext26d_32x4d | 1000 | 2605.98 |
| 16 | timm | seresnext26t_32x4d | 1000 | 2575.64 |
| 17 | timm | seresnext50_32x4d | 1000 | 4064.47 |
| 18 | timm | skresnet18 | 1000 | 1732.17 |
| 19 | timm | skresnet34 | 1000 | 3496.38 |
| 20 | timm | skresnext50_32x4d | 1000 | 4260.47 |
| 21 | timm | spnasnet_100 | 1000 | 318.08 |
| 22 | timm | ssl_resnet18 | 1000 | 1734.79 |
| 23 | timm | ssl_resnet50 | 1000 | 3921.04 |
| 24 | timm | ssl_resnext50_32x4d | 1000 | 4062.06 |
| 25 | timm | swin_base_patch4_window7_224 | 1000 | 102.33 |
| 26 | timm | swin_large_patch4_window7_224 | 1000 | 215.38 |
| 27 | timm | swin_small_patch4_window7_224 | 1000 | 61.28 |
| 28 | timm | swin_tiny_patch4_window7_224 | 1000 | 40.99 |
| 29 | timm | swsl_resnet18 | 1000 | 1734.79 |
| 30 | timm | swsl_resnet50 | 1000 | 3921.04 |
| 31 | timm | swsl_resnext50_32x4d | 1000 | 4062.06 |
| 32 | timm | tf_efficientnet_b0 | 1000 | 351.21 |
| 33 | timm | tf_efficientnet_b0_ap | 1000 | 351.21 |
| 34 | timm | tf_efficientnet_b0_ns | 1000 | 351.21 |
| 35 | timm | tf_efficientnet_cc_b0_4e | 1000 | 154.99 |
| 36 | timm | tf_efficientnet_cc_b0_8e | 1000 | 154.99 |
| 37 | timm | tf_efficientnet_es | 1000 | 1513.29 |
| 38 | timm | tf_efficientnet_lite0 | 1000 | 350.60 |
| 39 | timm | tf_mixnet_l | 1000 | 492.45 |
| 40 | timm | tf_mixnet_m | 1000 | 295.36 |
| 41 | timm | tf_mixnet_s | 1000 | 214.28 |
| 42 | timm | tf_mobilenetv3_large_075 | 1000 | 137.04 |
| 43 | timm | tf_mobilenetv3_large_100 | 1000 | 195.96 |
| 44 | timm | tf_mobilenetv3_large_minimal_100 | 1000 | 190.21 |
| 45 | timm | tf_mobilenetv3_small_075 | 1000 | 33.62 |
| 46 | timm | tf_mobilenetv3_small_100 | 1000 | 46.07 |
| 47 | timm | tf_mobilenetv3_small_minimal_100 | 1000 | 42.55 |
| 48 | timm | tnt_s_patch16_224 | 1000 | 33.18 |
| 49 | timm | tv_densenet121 | 1000 | 2703.08 |
| 50 | timm | tv_resnet34 | 1000 | 3501.19 |
| 51 | timm | tv_resnet50 | 1000 | 3921.04 |
| 52 | timm | tv_resnext50_32x4d | 1000 | 4062.06 |
| 53 | timm | twins_pcpvt_base | 1000 | 561.73 |
| 54 | timm | twins_pcpvt_large | 1000 | 794.99 |

| 55 | timm | twins_pcpvt_small | 1000 | 317.77 |
| 56 | timm | twins_svt_base | 1000 | 534.18 |
| 57 | timm | twins_svt_large | 1000 | 940.45 |
| 58 | timm | twins_svt_small | 1000 | 192.95 |
| 59 | timm | visformer_small | 1000 | 4532.08 |
| 60 | timm | vit_base_patch16_224 | 1000 | 192.20 |
| 61 | timm | vit_base_patch16_224_miil | 1000 | 192.17 |
| 62 | timm | vit_base_patch32_224 | 1000 | 192.09 |
| 63 | timm | cait_s24_224 | 1000 | 99.52 |
| 64 | timm | cait_xxs24_224 | 1000 | 38.79 |
| 65 | timm | cait_xxs36_224 | 1000 | 43.87 |
| 66 | timm | coat_lite_mini | 1000 | 135.35 |
| 67 | timm | coat_lite_small | 1000 | 170.24 |
| 68 | timm | coat_lite_tiny | 1000 | 107.48 |
| 69 | timm | coat_mini | 1000 | 284.68 |
| 70 | timm | coat_tiny | 1000 | 206.37 |
| 71 | timm | convit_base | 1000 | 192.18 |
| 72 | timm | convit_small | 1000 | 88.16 |
| 73 | timm | convit_tiny | 1000 | 32.85 |
| 74 | timm | cspresnext50 | 1000 | 2942.24 |
| 75 | timm | deit_base_distilled_patch16_224 | 1000 | 192.93 |
| 76 | timm | deit_base_patch16_224 | 1000 | 192.20 |
| 77 | timm | deit_small_distilled_patch16_224 | 1000 | 76.22 |
| 78 | timm | deit_small_patch16_224 | 1000 | 75.85 |
| 79 | timm | deit_tiny_distilled_patch16_224 | 1000 | 33.04 |
| 80 | timm | deit_tiny_patch16_224 | 1000 | 32.86 |
| 81 | timm | densenet121 | 1000 | 2703.08 |
| 82 | timm | densenet169 | 1000 | 3204.45 |
| 83 | timm | densenet201 | 1000 | 4092.84 |
| 84 | timm | densenetblur121d | 1000 | 2931.62 |
| 85 | timm | dla34 | 1000 | 2927.88 |
| 86 | timm | dla46_c | 1000 | 556.25 |
| 87 | timm | dla46x_c | 1000 | 518.84 |
| 88 | timm | dla60 | 1000 | 4059.07 |
| 89 | timm | dla60_res2net | 1000 | 3955.43 |
| 90 | timm | dla60_res2next | 1000 | 3323.87 |
| 91 | timm | dla60x | 1000 | 3380.01 |
| 92 | timm | dla60x_c | 1000 | 565.83 |
| 93 | timm | dm_nfnet_f1 | 1000 | 45.10 |
| 94 | timm | dpn68 | 1000 | 2218.44 |
| 95 | timm | dpn68b | 1000 | 2218.44 |
| 96 | timm | eca_nfnet_l0 | 1000 | 2.73 |
| 97 | timm | ecaresnet50d | 1000 | 4151.53 |
| 98 | timm | ecaresnet50d_pruned | 1000 | 2413.62 |
| 99 | timm | ecaresnet101d_pruned | 1000 | 3313.54 |
| 100 | timm | ecaresnetlight | 1000 | 3916.31 |
| 101 | timm | efficientnet_b0 | 1000 | 368.01 |
| 102 | timm | efficientnet_b1 | 1000 | 543.33 |
| 103 | timm | efficientnet_es | 1000 | 1705.92 |
| 104 | timm | efficientnet_es_pruned | 1000 | 1705.92 |
| 105 | timm | efficientnet_lite0 | 1000 | 367.40 |
| 106 | timm | ese_vovnet19b_dw | 1000 | 1271.76 |
| 107 | timm | fbnetc_100 | 1000 | 367.22 |
| 108 | timm | gernet_m | 1000 | 2865.44 |
| 109 | timm | gernet_s | 1000 | 709.75 |
| 110 | timm | ghostnet_100 | 1000 | 140.10 |
| 111 | timm | gluon_resnet18_v1b | 1000 | 1734.79 |
| 112 | timm | gluon_resnet34_v1b | 1000 | 3501.19 |
| 113 | timm | gluon_resnet50_v1b | 1000 | 3921.04 |

| 114 | timm | gluon_resnet50_v1c | 1000 | 4151.11 |
|---|---|---|---|---|
| 115 | timm | gluon_resnet50_v1d | 1000 | 4151.45 |
| 116 | timm | gluon_resnext50_32x4d | 1000 | 4062.06 |
| 117 | timm | gluon_seresnext50_32x4d | 1000 | 4064.47 |
| 118 | timm | gmixer_24_224 | 1000 | 78.45 |
| 119 | timm | gmlp_s16_224 | 1000 | 55.07 |
| 120 | timm | hardcorenas_a | 1000 | 212.67 |
| 121 | timm | hardcorenas_b | 1000 | 238.99 |
| 122 | timm | hardcorenas_c | 1000 | 256.37 |
| 123 | timm | hardcorenas_d | 1000 | 273.29 |
| 124 | timm | hardcorenas_e | 1000 | 321.83 |
| 125 | timm | hardcorenas_f | 1000 | 324.90 |
| 126 | timm | hrnet_w18 | 1000 | 4116.73 |
| 127 | timm | hrnet_w18_small | 1000 | 1538.74 |
| 128 | timm | hrnet_w18_small_v2 | 1000 | 2494.39 |
| 129 | timm | legacy_seresnet18 | 1000 | 1734.88 |
| 130 | timm | legacy_seresnet34 | 1000 | 3501.34 |
| 131 | timm | legacy_seresnet50 | 1000 | 3701.95 |
| 132 | timm | legacy_seresnext26_32x4d | 1000 | 2375.58 |
| 133 | timm | legacy_seresnext50_32x4d | 1000 | 4064.47 |
| 134 | timm | levit_128 | 1000 | 55.91 |
| 135 | timm | levit_128s | 1000 | 54.54 |
| 136 | timm | levit_192 | 1000 | 112.07 |
| 137 | timm | levit_256 | 1000 | 194.78 |
| 138 | timm | levit_384 | 1000 | 426.22 |
| 139 | timm | mixer_b16_224 | 1000 | 166.90 |
| 140 | timm | mixer_b16_224_miil | 1000 | 166.90 |
| 141 | timm | mixer_l16_224 | 1000 | 344.89 |
| 142 | timm | mixnet_l | 1000 | 529.61 |
| 143 | timm | mixnet_m | 1000 | 323.54 |
| 144 | timm | mixnet_s | 1000 | 228.17 |
| 145 | timm | mixnet_xl | 1000 | 862.22 |
| 146 | timm | mnasnet_100 | 1000 | 299.91 |
| 147 | timm | mobilenetv2_100 | 1000 | 286.90 |
| 148 | timm | mobilenetv2_110d | 1000 | 410.63 |
| 149 | timm | mobilenetv2_120d | 1000 | 637.93 |
| 150 | timm | mobilenetv2_140 | 1000 | 555.31 |
| 151 | timm | mobilenetv3_large_100 | 1000 | 205.38 |
| 152 | timm | mobilenetv3_large_100_miil | 1000 | 205.38 |
| 153 | timm | mobilenetv3_rw | 1000 | 205.38 |
| 154 | timm | nfnet_l0 | 1000 | 13.09 |
| 155 | timm | pit_b_224 | 1000 | 209.98 |
| 156 | timm | pit_b_distilled_224 | 1000 | 210.96 |
| 157 | timm | pit_s_224 | 1000 | 99.93 |
| 158 | timm | pit_s_distilled_224 | 1000 | 100.48 |
| 159 | timm | pit_ti_224 | 1000 | 39.09 |
| 160 | timm | pit_ti_distilled_224 | 1000 | 39.34 |
| 161 | timm | pit_xs_224 | 1000 | 61.83 |
| 162 | timm | pit_xs_distilled_224 | 1000 | 62.20 |
| 163 | timm | regnetx_002 | 1000 | 189.83 |
| 164 | timm | regnetx_004 | 1000 | 379.27 |
| 165 | timm | regnetx_006 | 1000 | 573.07 |
| 166 | timm | regnetx_008 | 1000 | 762.68 |
| 167 | timm | regnetx_016 | 1000 | 1528.64 |
| 168 | timm | regnetx_032 | 1000 | 3029.51 |
| 169 | timm | regnetx_040 | 1000 | 3780.90 |
| 170 | timm | regnety_002 | 1000 | 190.29 |
| 171 | timm | regnety_004 | 1000 | 383.25 |
| 172 | timm | regnety_006 | 1000 | 573.40 |

| 173 | timm | regnety_008 | 1000 | 760.28 |
| 174 | timm | regnety_016 | 1000 | 1537.44 |
| 175 | timm | regnety_032 | 1000 | 3029.41 |
| 176 | timm | regnety_040 | 1000 | 3789.28 |
| 177 | timm | repvgg_b0 | 1000 | 3239.87 |
| 178 | timm | res2net50_14w_8s | 1000 | 4010.80 |
| 179 | timm | res2net50_26w_4s | 1000 | 4082.33 |
| 180 | timm | res2net50_48w_2s | 1000 | 3989.85 |
| 181 | timm | res2next50 | 1000 | 4005.80 |
| 182 | timm | resmlp_12_224 | 1000 | 69.52 |
| 183 | timm | resmlp_12_distilled_224 | 1000 | 69.52 |
| 184 | timm | resmlp_24_224 | 1000 | 83.49 |
| 185 | timm | resmlp_24_distilled_224 | 1000 | 83.49 |
| 186 | timm | resmlp_36_224 | 1000 | 97.45 |
| 187 | timm | resmlp_36_distilled_224 | 1000 | 97.45 |
| 188 | timm | resmlp_big_24_224 | 1000 | 233.73 |
| 189 | timm | resmlp_big_24_224_in22ft1k | 1000 | 233.73 |
| 190 | timm | resmlp_big_24_distilled_224 | 1000 | 233.73 |
| 191 | timm | resnest14d | 1000 | 2636.61 |
| 192 | timm | resnest26d | 1000 | 3475.13 |
| 193 | timm | resnest50d_1s4x24d | 1000 | 4222.56 |
| 194 | timm | resnest50d_4s2x40d | 1000 | 4201.50 |
| 195 | timm | resnet18 | 1000 | 1734.79 |
| 196 | timm | resnet18d | 1000 | 1964.95 |
| 197 | timm | resnet26 | 1000 | 2247.86 |
| 198 | timm | resnet26d | 1000 | 2478.27 |
| 199 | timm | resnet34 | 1000 | 3501.19 |
| 200 | timm | resnet34d | 1000 | 3731.34 |
| 201 | timm | resnet50 | 1000 | 3921.04 |
| 202 | timm | resnet50d | 1000 | 4151.45 |
| 203 | timm | resnetblur50 | 1000 | 4914.30 |
| 204 | timm | resnetv2_50x1_bit_distilled | 1000 | 2.05 |
| 205 | timm | vit_large_patch16_224 | 1000 | 436.37 |
| 206 | timm | vit_large_r50_s32_224 | 1000 | 387.23 |
| 207 | timm | vit_small_patch16_224 | 1000 | 75.85 |
| 208 | timm | vit_small_patch32_224 | 1000 | 75.79 |
| 209 | timm | vit_small_r26_s32_224 | 1000 | 57.42 |
| 210 | timm | vit_tiny_patch16_224 | 1000 | 32.86 |
| 211 | timm | vit_tiny_r_s16_p8_224 | 1000 | 42.02 |
| 212 | pretrainedmodels | alexnet | 1000 | 681.57 |
| 213 | pretrainedmodels | densenet121 | 1000 | 2732.91 |
| 214 | pretrainedmodels | densenet169 | 1000 | 3240.65 |
| 215 | pretrainedmodels | densenet201 | 1000 | 4139.87 |
| 216 | pretrainedmodels | dpn68 | 1000 | 2241.70 |
| 217 | pretrainedmodels | nasnetamobile | 1000 | 551.83 |
| 218 | pretrainedmodels | resnet18 | 1000 | 1734.79 |
| 219 | pretrainedmodels | resnet34 | 1000 | 3501.19 |
| 220 | pretrainedmodels | resnet50 | 1000 | 3921.04 |
| 221 | pretrainedmodels | se_resnet50 | 1000 | 3707.13 |
| 222 | swag | regnety_16gf_in1k | 1000 | 15223.83 |
| 223 | swag | regnety_32gf_in1k | 1000 | 30849.35 |
| 224 | timm | beit_base_patch16_224 | 1000 | 171.92 |
| 225 | timm | beit_large_patch16_224 | 1000 | 364.30 |
| 226 | timm | convnext_base | 1000 | 616.58 |
| 227 | timm | convnext_base_384_in22ft1k | 1000 | 616.58 |
| 228 | timm | convnext_base_in22ft1k | 1000 | 616.58 |
| 229 | timm | convnext_large | 1000 | 1205.29 |
| 230 | timm | convnext_large_384_in22ft1k | 1000 | 1205.29 |
| 231 | timm | convnext_large_in22ft1k | 1000 | 1205.29 |

| 232 | timm | convnext_nano | 1000 | 2340.25 |
|---|---|---|---|---|
| 233 | timm | convnext_nano_hnf | 1000 | 2340.25 |
| 234 | timm | convnext_nano_ols | 1000 | 2386.90 |
| 235 | timm | convnext_small | 1000 | 392.34 |
| 236 | timm | convnext_small_384_in22ft1k | 1000 | 392.34 |
| 237 | timm | convnext_small_in22ft1k | 1000 | 392.34 |
| 238 | timm | convnext_tiny | 1000 | 307.45 |
| 239 | timm | convnext_tiny_384_in22ft1k | 1000 | 307.45 |
| 240 | timm | convnext_tiny_hnf | 1000 | 4261.97 |
| 241 | timm | convnext_tiny_in22ft1k | 1000 | 307.45 |
| 242 | timm | convnext_xlarge_384_in22ft1k | 1000 | 1980.92 |
| 243 | timm | convnext_xlarge_in22ft1k | 1000 | 1980.92 |
| 244 | MAE | vit_base_patch16 | 1000 | 192.21 |
| 245 | MAE | vit_large_patch16 | 1000 | 436.38 |
| 246 | MAE | vit_huge_patch14 | 1000 | 785.64 |
| 247 | swag | vit_b16 | 768 | 164.43 |
| 248 | swag | vit_l16 | 1024 | 339.30 |
| 249 | swag | vit_h14 | 1280 | 584.25 |
| 250 | swag | regnety_16gf | 3024 | 15220.81 |
| 251 | swag | regnety_32gf | 3712 | 30845.63 |
| 252 | timm | vit_base_patch16_224_miil_in21k | 11221 | 199.67 |
| 253 | timm | mixer_b16_224_miil_in21k | 11221 | 174.39 |
| 254 | timm | mobilenetv3_large_100_miil_in21k | 11221 | 205.38 |
| 255 | timm | swin_base_patch4_window7_224_in22k | 21841 | 122.71 |
| 256 | timm | swin_large_patch4_window7_224_in22k | 21841 | 245.93 |
| 257 | timm | beit_base_patch16_224_in22k | 21841 | 187.21 |
| 258 | timm | beit_large_patch16_224_in22k | 21841 | 384.68 |
| 259 | timm | convnext_base_in22k | 21841 | 636.96 |
| 260 | timm | convnext_large_in22k | 21841 | 1235.83 |
| 261 | timm | convnext_small_in22k | 21841 | 407.62 |
| 262 | timm | convnext_tiny_in22k | 21841 | 322.74 |
| 263 | timm | convnext_xlarge_in22k | 21841 | 2021.65 |
| 264 | timm | resnetv2_50x1_bitm_in21k | 21843 | 42.78 |
| 265 | timm | resnetv2_50x3_bitm_in21k | 21843 | 128.30 |
| 266 | timm | resnetv2_101x1_bitm_in21k | 21843 | 42.78 |
| 267 | timm | resnetv2_152x2_bitm_in21k | 21843 | 85.54 |
| 268 | timm | vit_base_patch16_224_in21k | 21843 | 207.49 |
| 269 | timm | vit_base_patch32_224_in21k | 21843 | 207.38 |
| 270 | timm | vit_base_r50_s16_224_in21k | 21843 | 244.24 |
| 271 | timm | mixer_b16_224_in21k | 21843 | 182.18 |
| 272 | timm | mixer_l16_224_in21k | 21843 | 365.27 |
| 273 | timm | vit_huge_patch14_224_in21k | 21843 | 811.09 |
| 274 | timm | vit_large_patch16_224_in21k | 21843 | 456.75 |
| 275 | timm | vit_large_patch32_224_in21k | 21843 | 456.61 |
| 276 | timm | vit_large_r50_s32_224_in21k | 21843 | 407.61 |
| 277 | timm | vit_small_patch16_224_in21k | 21843 | 83.50 |
| 278 | timm | vit_small_patch32_224_in21k | 21843 | 83.45 |
| 279 | timm | vit_small_r26_s32_224_in21k | 21843 | 65.07 |
| 280 | timm | vit_tiny_patch16_224_in21k | 21843 | 36.70 |
| 281 | timm | vit_tiny_r_s16_p8_224_in21k | 21843 | 45.86 |
| 282 | clip | ViT-B-16 | arbitrary | 188.32 |
| 283 | clip | ViT-B-32 | arbitrary | 188.32 |

