# OpenReview forum: "SIMPLE: Specialized Model-Sample Matching for Domain Generalization"
_ICLR.cc/2023/Conference — ICLR 2023 poster_

### Official Review · Reviewer_KF3m · 2022-10-23

**Confidence:** 5
**Correctness:** 4
**Technical Novelty And Significance:** 3
**Empirical Novelty And Significance:** 4
**Recommendation:** 8

**Clarity, Quality, Novelty And Reproducibility:**

The approach of using a set of pre-trained models for domain generalization has been suggested before. See Balaji, et al, NeurIPS 2018. The difference, which I think is important, is how many pre-trained models are assumed to be available. It will be good to come up with a strategy for retaining the most useful pre-trained models.
The paper has ICLR quality, in that the theoretical basis is strong.
Reproducibility will be challenging given the number of pre-trained models used.

**Strength And Weaknesses:**

Strengths:  An approach supported by theoretical analysis and empirical evidence based on extensive evaluations for domain generalization that assumes the availability of a large number of pre-trained source models. The success of domain generalization depends on how close the one or more pre-trained models are in terms of addressing the domain shift. Extensive experimental results and ablation studies are included.
Weaknesses: This approach  reminds me of a NeurIPS paper by Y. Balaji, S. Sankaranarayanan and R. Chellappa, “MetaReg: Towards Domain Generalization Using Meta-regularization”, Proc. Neural and Information Processing Systems, Montreal, Dec. 2018. There is a brute-force philosophy embedded in the approach. Even more performance gains can be achieved if one can assume the availability of 500 or 1000 or 10,000 models!

**Summary Of The Paper:**

Domain generalization is a challenging problem as target data is not concurrently available along with source data. One approach is to have a set of pre-trained source models and develop a strategy for domain generalization. A paper by Balaji et al., (NeurIPS 2018) suggested the approach of using a set of pre-trained models and set the problem of domain generalization as a regularization problem. This paper has taken a similar approach but assumes the availability of a large number of pre-trained models. The domain generalization problem is then solved y dispatching the pre-trained models to OOD samples based on their matching metric to the target task. Extensive evaluations show that the proposed approach has better generalization performance and significantly higher training efficiency compared to existing DG methods.

**Summary Of The Review:**

Domain generalization is harder than domain adaptation. The proposed approach of leveraging a large set of pre-trained models to help with generalization is a good idea. One can see this approach as an extension of nearest neighbor rule, in which the more training data you have, the better the classification performance is. In the proposed approach, each model can be viewed as a training sample. This makes me wonder if some asymptotic performance results can be derived as is done with nearest neighbor rules. Discuss what happens if you assume a very very large number of pre-trained models.

---

> ### Author Response · Authors · 2022-11-14
> **Response to Reviewer KF3m**
>
> Thank you for the recognition of our work from the perspective of novelty, paper quality, and technical contribution. And we also appreciate your positive feedback and suggestions for improvement of our paper!
>
> > **Q1. Missing related work (MetaReg).**
>
> Thank you for pointing out this work. We have revised the related work part of the paper in Appendix A.1 accordingly.
>
> MetaReg [1] learns a regularizer at a meta-level to make model trained on source domains to generalize to target domains.
> It proposes similar idea of leveraging pretrained model as a fixed feature extractor to handle domain generalization problem.
> The differences of our method SIMPLE to this paper lie in that: (1) SIMPLE leverages significantly diverse pretrained models to explore their generalization ability across unseen domains; (2) instead of training individual linear classifiers for a pretrained model, SIMPLE trains a shared label space adapter for efficient adaption of these pretrained models.
>
> ---
>
> > **Q2. It will be good to come up with a strategy for retraining the most useful pre-trained models.**
>
> Thank you for the constructive suggestion and we have come up with a simple strategy for this aim.
> One promising direction is to retrain the models with low generalization ability (i.e., poor transferring performance from the pretrained domain to the unseen domains during training) to remedy and improve their performance, according to the output of our dispatcher network.
>
> Note that, as an initial proof-of-concept exploration for taking model-sample matching to tackle domain generalization problem, we only adopt fixed pretrained models based on the consideration of efficiency. However, we believe that fine-tuning *promising* models to improve their generalization ability is certainly an interesting direction.
>
> Nevertheless, retraining the pretrained models while keeping their generalization ability is non-trivial. For example, the recent work [2] found that fine-tuning can actually distort pretrained features, resulting in worse generalization performance than linear probing.
> Thus, we leave it as a future exploration.
>
> ---
>
> > **Q3. Reproducibility will be challenging given the number of pre-trained models used.**
>
> We acknowledge that the size of the model pool used in our method SIMPLE could become large. However, based on this, SIMPLE can match best-performing models for different distributions.
> But note that the reproducibility is not so difficult as:
>
> (1) Constructing the model pool is of low cost. Note that the process of obtaining these pretrained models is virtually like downloading papers, given various publicly available collections (e.g., [timm](https://github.com/rwightman/pytorch-image-models)) of models.
>
> (2) Moreover, the computation cost and memory cost would not be linearly increased with the model pool size, but remains low, as explained in response to Q2 of Reviewer qJrT.
>
> And we will release our code and reproducible execution scripts upon acceptance to facilitate reproduction.
>
> ---
>
> > **Q4. If some asymptotic performance results can be derived as is done with nearest neighbor rules. Discuss what happens if you assume a very very large number of pre-trained models.**
>
> Very insightful interpretation of the effectiveness of SIMPLE from the perspective of the nearest neighbor rules [3].
>
> These rules suggest that asymptotically optimal performance can be achieved when the number of pre-classified samples is very large. Different from classic works, SIMPLE incorporates diverse pretrained models instead of samples and matches pretrained models with samples. Currently, with 283 pretrained models, SIMPLE has established promising generalization performance. If the model pool size goes to infinity, the Bayes' risk might be achieved for all real-world samples!
>
> ---
>
> References:\
> [1] Balaji, Yogesh, et al. "Metareg: Towards domain generalization using meta-regularization." Advances in neural information processing systems 31 (2018).\
> [2] Kumar, Ananya, et al. "Fine-tuning can distort pretrained features and underperform out-of-distribution." ICLR 2022.\
> [3] Wilson, Dennis L. "Asymptotic properties of nearest neighbor rules using edited data." IEEE Transactions on Systems, Man, and Cybernetics 3 (1972): 408-421.

---

> > ### Author Response · Authors · 2022-11-18
> > **Sincerely Looking Forward to Your Further Suggestions**
> >
> > Dear Reviewer,
> >
> > Given that the discussion period is coming to end soon, we sincerely look forward to your reply to our response, and we are open to any discussion to improve our paper.
> >
> > Best wishes,
> >
> > The authors.

---

### Official Review · Reviewer_At6g · 2022-10-23

**Confidence:** 4
**Correctness:** 3
**Technical Novelty And Significance:** 2
**Empirical Novelty And Significance:** 4
**Recommendation:** 6

**Clarity, Quality, Novelty And Reproducibility:**

The paper is novel (to my knowledge) and original (except theorems), but not entirely clear. I think the authors should invest in more clear writing in the paper because some statements are very confusing.

**Strength And Weaknesses:**

**Strengths:**

The authors propose a creative way how to use a pool of pretrained models to achieve better generalization on distribution shifts. They demonstrate SOTA results, which significantly outperform previous models. They demonstrate the necessity of parts of their algorithm via an ablation study.

**Weaknesses and comments:**

 - The writing is a bit confusing. First, the authors say that they *“adopt a simple la- bel adapter that projects the label space of the pretrained domain to that of the target domain”*, but then they say *“with the adapter training on source domains”*. After reading several times, I understand what they meant in the beginning, but upon first reading, it was very confusing. The authors should stick to the strict terminology that they introduce in the very beginning, e.g. “pre-training domain” (the one that corresponds to ImageNet) and then “source” and “target” domains from a typical domain generalization literature.
- No free lunch in domain generalization was demonstrated in several works before, e.g. in the DomainBed paper that the authors cite. So Takeaway 2 is an extended version of those results, but with varying architectures. It is even less surprising, given the fact that the authors are not trying to address distribution shift at all there.
- For me, it is not obvious why Theorem 1 would explicitly generalize to neural networks of fixed size.
- Also by just tuning the label on the source domain (not the same as image-net pretraining), it is not obvious that the model can even achieve minimal test risk and if all theoretical justifications would be true. An arbitrary model trained under ERM can exhibit quite arbitrary behavior under distribution shift, as known from many previous theoretical results. By only allowing to tune the last layer, the authors put quite a strong constraint on the class of models under which optimal risk can be achieved, therefore such models are not guaranteed to generalize.
- Remark 3 is also not clear to me. Any causal model would achieve optimal risk under the assumption of covariate shift.
-  How *“fitness of the pretrained models to the target distributions”* is evaluated?
- *“Table 1. Baseline results are from original papers with the same setup”*. Does it mean that the method’s performance for benchmarks is demonstrated on “one” model (originally proposed in the corresponding paper). So it is possible that the best architecture which would allow achieving a higher number in this table might have not been considered? My concern is that majority of the papers were focused on demonstrating the effect of the *learning method given a fixed architecture* and that with another architecture, the same learning method would achieve another number. So comparing it to the pool of > 200 models is not entirely fair.
- Ablation study RQ2: the averaging strategy indeed looks not optimal. But why the “averaging” weights can’t be learned by backpropagation?


**Summary Of The Paper:**

The paper proposes to leverage a pool of pretrained models without expensive fine-tuning. They empirically demonstrate the proposed method achieves SOTA results on several benchmarks without significant loss in inference speed.


**Summary Of The Review:**

From the way the paper is written, many aspects are not clear to me, which I stated in the section above. Overall the paper looks quite interesting and the empirical evaluation of the paper looks significant. However, I don’t know how to properly evaluate this paper when there are so many unclear things in it. In particular, the theoretical part and big parts of the paper are very confusing, as well as the model description. I am willing to change my evaluation when my questions are clarified.

---

> ### Author Response · Authors · 2022-11-14
> **Response to Reviewer At6g (part 3)**
>
> > **Q8. Ablation study RQ2: Why the “averaging” weights can’t be learned by backpropagation?**
>
> In RQ2, we compare our method SIMPLE (with the model dispatcher learning model specialty at the meta-level) with the random ensemble [3] method (using unlearnable ensemble weights) to investigate the necessity of specialty learning.\
> We further clarify the new setting you mentioned and compare SIMPLE in the newly conducted experiment.
>
> #### **Clarification**
>
> For ensemble-based approaches, the overall prediction is given by
> $$
> \hat{y}_i = \boldsymbol{w}^{\top} [f_1(\boldsymbol{x}_i), \cdots, f_K(\boldsymbol{x}_i)],
> $$
> where $w_k$ is the weight for aggregating the prediction of $k$-th models $f_k(\boldsymbol{x}_i)$.
>
> In SIMPLE, the ensemble weights are given by $\boldsymbol{w} = \text{MLP}(\boldsymbol{c}^{\top}_i \boldsymbol{C})$, where $\boldsymbol{c}_i$ is the embedding for the $i$-th sample and $\boldsymbol{C}$ contains the embeddings for all the models.
>
> Therefore, "learn the averaging weights by backpropagation" is a special case of proposed ensemble approach, where the ensemble weights $\boldsymbol{w} \in \mathbb{R}^K$ are randomly initialized and optimized through back-propagation. Here in this simplified version, $\boldsymbol{w}$ is shared for all the samples in the dataset and does not incorporate model or sample information.\
> Compared with this simplified version, SIMPLE explicitly leverages sample and model information (encoded in the embedding vectors $\boldsymbol{c}_i$ and $\boldsymbol{C}$) to generate specialized ensemble weights for each sample, which is more fine-grained. With model and sample embedding, SIMPLE enjoys much lower training cost when incorporating new pretrained models not in the model pool.
>
> #### **New Experiment**
>
> We further implement this special case and conduct experiments on OfficeHome dataset (due to time limit) to compare with SIMPLE.
>
> |      Model Pool-B      | OfficeHome |
> |------------------------|------------|
> | _directly learned ensemble weights_ | _82.3_     |
> | existing SOTA (EoA)    | 83.9       |
> | SIMPLE                 | **87.7**   |
>
> By performing sufficient hyperparameter tuning, such a simple version achieves an average accuracy of 82.3% on OfficeHome, which is worse than 87.7% of SIMPLE (and even worse than 84.6% of SIMPLE using the much smaller Model Pool-A) and 83.9% of the existing SOTA, as shown in the above table.
> It in contrast suggests that, incorporating information about models and the sample to conduct fine-grained model-sample matching in SIMPLE, is necessary and more effective, than using a set of weights and optimized by back-propagation.
> Due to time limit, we cannot finish experiments on all datasets and will add this ablation study on all datasets in the future version of the paper.
>
> Thanks for pointing this out, we have added this experiment in Appendix A.8.
>
> ---
>
> Reference:\
> [1] Kumar, Ananya, et al. "Fine-tuning can distort pretrained features and underperform out-of-distribution." ICLR 2022.\
> [2] Wortsman, Mitchell, et al. "Robust fine-tuning of zero-shot models." CVPR 2022.\
> [3] Lakshminarayanan, Balaji, Alexander Pritzel, and Charles Blundell. "Simple and scalable predictive uncertainty estimation using deep ensembles." NeurIPS 2017.

---

> > ### Comment · Reviewer_At6g · 2022-12-05
> > **thank you to the authors for detailed answers to the questions**
> >
> > I would like to thank the authors for the clear and detailed answers. I updated my score accordingly.

---

> > > ### Author Response · Authors · 2022-12-08
> > > **Thank you for your reply!**
> > >
> > > Dear Reviewer At6g,
> > >
> > > Thank you for the response. We are glad our responses were helpful in addressing your concerns. We really appreciate your time and insightful comments that led to improvement of our manuscript.
> > >
> > > Best wishes,
> > >
> > > The authors.

---

> ### Author Response · Authors · 2022-11-14
> **Response to Reviewer At6g (part 2)**
>
> > **Q5. "Remark 3 is not clear to me. Any causal model would achieve optimal risk under the assumption of covariate shift."**
>
> We agree with the reviewer that a causal model can achieve the optimal risk *averaged over all unseen domains*. However, Remark 3 considers a specific domain. Due to the presence of noise, a causal model cannot achieve zero risk w.r.t. one specific domain. The domain-specific features are correlated with the labels and the pretrained models exploiting these features have lower risks.
>
> Therefore, a causal model achieves the optimal risk when restricting only one model to be utilized. However, when more models are available, we can dispatch the models to their most suitable domains, which is able to achieve lower risks.
>
> Motivated by this evidence, our method SIMPLE incorporates a model pool consisting of diverse models and learns to *dispatch varied best-matching models across different domains/samples*. By experiments, our method can perform better than invariant causal prediction-based methods (e.g., IRM).
>
> ---
>
> > **Q6. How “fitness of the pretrained models to the target distributions” is evaluated?**
>
> “Fitness of the pretrained models to the *target* distributions” is only measured in the test phase.
>
> Generally, our method can measure the fitness between the pretrained models and the ***unseen*** distribution, by learning the matching metric between the pretrained model to the unseen samples in the source domain.
>
> Specifically, as explained in Q1 of the general response part above, both "source domains" and "target domains" are unseen for the pretrained models in the "pretraining domains".\
> During training, our method handles the distribution shifts for the pretrained models by training the dispatcher network on the source domains upon the fixed pretrained models.\
> So, in the test phase, when given one sample from the target domain, the trained dispatcher network in our method can leverage the matching metric between the pretrained models to this test sample just like that in the training phase.
>
> ---
>
> > **Q7. comparing baselines to the pool of > 200 models is not entirely fair.**
>
> We acknowledge that the number of models may seem large, but the comparison is certainly fair given that:
>
> (1) Our method SIMPLE involves significantly fewer learnable parameters. Though adopting various pretrained models, SIMPLE involves significantly fewer learnable parameters (only the lightweight model dispatcher and label space adapters) than baselines that need to fine-tune large-scale pretrained models.
>
> (2) Constructing and using the model pool is of low cost. Note that the process of obtaining these pretrained models is virtually of very low cost, given various publicly available collections (e.g., [timm](https://github.com/rwightman/pytorch-image-models)) of models. Moreover, the computation cost would not be linearly increased with the model pool size, but remains low, as explained in response to Q2 of Reviewer qJrT.
>
> (3) Only limited models are dispatched for each domain/sample by SIMPLE. First, model dispatcher only activates a few appropriate models (6 in our setting) for the inference of each sample. Second, in our newly conducted experiments below, we show that we can optimize the quantity of model pool by utilizing the matching preference of model dispatcher (for more details can refer to response to Q5 of Reviewer ySiZ or Appendix A.12). By doing so, with only two models to compose the pool, SIMPLE can outperform SOTA on PACS and OfficeHome datasets, with results shown in following figures and tables.
>
> **Result of PACS dataset.**
>
> Figure: https://user-images.githubusercontent.com/17319095/201453468-dd55b3e2-ec2a-4d30-b6ce-64aef4a6e41d.png
>
> | Model pool size |   1   | Previous _SOTA_ |   2   |   10  |   30  |   50  |  100  |  150  |  200  |  250  |  283  |
> |:-----:|:-----:|:-----:|:-----:|:-----:|:-----:|:-----:|:-----:|:-----:|:-----:|:-----:|:-----:|
> | SIMPLE          | 95.60 | _96.80_ | 98.42 | 98.60 | 98.74 | 98.80 | 98.93 | 98.97 | 98.98 | 99.00 | 99.02 |
>
> **Result of OfficeHome dataset.**
>
> Figure: https://user-images.githubusercontent.com/17319095/201453374-414a1756-2177-4d5a-90cb-baa31ccdebb9.png
>
> | Model pool size |   1   | Previous _SOTA_ |   2   |   10  |   30  |   50  |  100  |  150  |  200  |  250  |  283  |
> |:-----:|:-----:|:-----:|:-----:|:-----:|:-----:|:-----:|:-----:|:-----:|:-----:|:-----:|:-----:|
> | SIMPLE          | 78.2 | _83.9_ | 84.94 | 85.72 | 85.93 | 88.26 | 88.17 | 88.05 | 87.94 | 87.87 | 87.74 |
>
> In summary, the (easily constructed and computationally efficient) model pool allows us to match the appropriate models for different distributions, and based on that even a very light model pool can deliver promising generalization ability.
>
> ---

---

> ### Author Response · Authors · 2022-11-14
> **Response to Reviewer At6g (part 1)**
>
> We thank the reviewer for the valuable comments! We have revised the paper to the confusion (Q1) and incorporated your suggestions, and we hope our response addresses your concerns.
>
> > **Q1: The writing is a bit confusing. The authors should stick to the strict terminology that they introduce in the very beginning, e.g. “pre-training domain” (the one that corresponds to ImageNet) and then “source” and “target” domains from a typical domain generalization literature.**
>
> We appreciate your valuable suggestion and we have carefully revised the manuscript to eliminate potential misunderstandings.
> And we also claim this in Q1 of our general response.
>
> ---
>
> > **Q2. Takeaway 2 (No dominant pretrained models across unseen domains) in Section 2 seems less surprising.**
>
> We acknowledge that there are similar conclusions drawn in related studies about domain generalization.
> However, in Takeaway 2, we focus on investigating the No Free Lunch Phenomenon over abundant pretrained models in terms of generalization, conducting large-scale experiments over 283 pretrained models with diverse network architectures, learning objectives, and pretraining domains, which has not been well studied in other literature.
>
> More importantly, this empirical evidence of there is no dominant pretrained model across varied domains/samples motivates us to propose our method, that is, exploiting the generalization ability of pretrained models appropriately to handle distributional shifts.
>
> The main contribution is that our method SIMPLE handles the distributional shifts with significantly lower adaption cost of pretrained models than existing domain generalization methods with fine-tuning.
> And the extensive empirical results illustrated that our method significantly outperforms the strong baselines.
>
> ---
>
> > **Q3. "Why Theorem 1 would explicitly generalize to neural networks of fixed size?"**
>
> Thanks for pointing out this important issue that requires more clear explanation in the manuscript, which results in confusion.
>
> Theorem 1 analyzes the OOD generalization error of kernel method. The kernel method consists of a static nonlinear feature extractor and a learnable linear classifier. Note that, the pretrained models are kept frozen without fine-tuning, which play the role of static feature extractors for the subsequent trainable linear classifier (the linear domain label adapter) in our method.
> Under such circumstance, the pretrained model acts as a kernel and thus Theorem 1 applies.
>
> Thanks for pointing this out, we have revised Remark 1 in the paper accordingly.
>
> ---
>
> > **Q4. "By only allowing to tune the last layer, the authors put quite a strong constraint on the class of models under which optimal risk can be achieved, therefore such models are not guaranteed to generalize."**
>
> We thank the reviewer for pointing out this interesting question.
>
> Recent studies [1,2] have found that fine-tuning the whole model can hurt generalization ability of a pretrained model, while only tuning the last layer may have lower OOD error. Therefore, fixing the weights of the feature extractor may not have a negative influence on the performance. Our method exploits this finding and greatly outperforms fine-tuning based approaches in experiments, with thousands of speedups in terms of training time.
>
> ---

---

> ### Author Response · Authors · 2022-11-18
> **Sincerely Looking Forward to Your Suggestions**
>
> Dear Reviewer,
>
> Given that the discussion period is coming to end soon, we sincerely look forward to your reply to our response, and we are open to any discussion to improve our paper.
>
> Best wishes,
>
> The authors.

---

### Official Review · Reviewer_ySiZ · 2022-10-24

**Confidence:** 3
**Correctness:** 2
**Technical Novelty And Significance:** 2
**Empirical Novelty And Significance:** 2
**Recommendation:** 5

**Clarity, Quality, Novelty And Reproducibility:**

Clarity:

1. The term $a$ is undefined in that theorem.
2. How to estimate $E^{matched}_g$, since in a DG framework we do not have any access to the target domain?
3. How many pre-trained models are enough to run the algorithm? How to optimize that quantity?

Quality: Though the experimental results of the paper are quite impressive, I think that the technical contribution of the paper is limited based on my comments about its weaknesses.


Novelty and reproducibility: The idea of efficiently using pre-trained models in DG is quite novel but maybe not very practical due to the computational cost. Honestly, I did not have time to verify the code, but I think the experimental results of the paper are reproducible.

**Strength And Weaknesses:**

[Strengths]: The proposed method is relatively simple and easy to implement. The experimental results are convincing and partly support the main claims of the paper.

[Weakness]: My major concern is about the theoretical contribution of the paper. I did not have time to verify the results in Theorem 1, but even if they are all correct, I could not find the role of pre-trained models in the generalization error in a DG. Furthermore, the idea of (pre-trained) model-sample matching in the proposed method may be impractical since in DG we are not allowed to have access to the samples in the target domain.

**Summary Of The Paper:**

The paper investigates the generalization performance of pretrained models on domain generalization or data distribution shift problems, leading to the ''no free lunch hypothesis'' of pre-trained models in domain generalization settings.  The authors then propose a new DG method that aims to directly leverages pretrained models without fine-tuning. The experimental results on the mainstream benchmark show that the proposed method outperforms other strong baselines.

**Summary Of The Review:**

Overall, I personally think that this submission is below the acceptance rate for an ICLR paper. I would suggest the authors focus more on the theoretical part to make it stronger.

---

> ### Author Response · Authors · 2022-11-14
> **Response to Reviewer ySiZ (part 2)**
>
> > **Q4. (mentioned in `Clarity`) "How to estimate $E^{\text{matched}}_g$, since in a DG framework we do not have any access to the target domain?"**
>
> The $E^{\text{matched}}_g$ in Theorem 1 indicates the generalization error when the *training distributions and test distributions are matched* (as explained in Section 2.2), i.e., it is the in-distribution error (error in the pretraining domain in our case) which is not relevant to unseen target domains.
> Thanks for pointing out this, we have clarified that in Section 2.2 and Appendix A.5.
>
> ---
>
> > **Q5. (mentioned in `Clarity`) "How many pre-trained models are enough to run the algorithm? How to optimize that quantity?"**
>
> Based on the theoretical evidence of *"generalization error relies on the fitness between the model and target distribution"*, the quantity of model pool is related to distributions of unseen target domains.
>
> Therefore, we construct a model pool with diverse pretrained models to cover the wide variety of distributions.
> Note that the process of obtaining these pretrained models is virtually of very low cost, given various publicly available collections (e.g., [timm](https://github.com/rwightman/pytorch-image-models)) of models.
> We do not hypothesize pretrained model selection, yet leave it to the model dispatcher, which has illustrated huge gains.
>
> Moreover, the computation cost would not be linearly increased with the model pool size, but remains low, as explained in response to Q2 of Reviewer qJrT.
>
> Since the model dispatcher of our method SIMPLE learns to match the best suitable models for unseen samples at a meta-level, its matching preference over unseen domains (*without* accessing ground truth labels) can be regarded as a measurement of model transferability on these unseen samples.
>
> Therefore, we can optimize the quantity of the model pool in a two-stage manner:
>
> (1) first learn with a large model pool on source domains to capture and calculate matching preferences;
>
> (2) then reconstruct a smaller model pool to include models preferred by our dispatcher.
>
> #### **New Experiment**
>
> **Experimental setup.** We experiment this two-stage manner training on PACS and OfficeHome datasets, using different reconstructed model pool sizes in the second stage.
>
> **Experimental results.** The results are shown in the uploaded anonymous figures and the following tables.
> As can be seen, given that SIMPLE can automatically match models more suitable for transferring to unseen domains, we can actually go beyond the existing SOTA approach even using a model pool with only two models (*while the best single model fails to outperform since there is No Free Lunch*).
>
> **Result of PACS dataset.**
>
> Figure: https://user-images.githubusercontent.com/17319095/201453468-dd55b3e2-ec2a-4d30-b6ce-64aef4a6e41d.png
>
> | Model pool size |   1   | Previous _SOTA_ |   2   |   10  |   30  |   50  |  100  |  150  |  200  |  250  |  283  |
> |:-----:|:-----:|:-----:|:-----:|:-----:|:-----:|:-----:|:-----:|:-----:|:-----:|:-----:|:-----:|
> | SIMPLE          | 95.60 | _96.80_ | 98.42 | 98.60 | 98.74 | 98.80 | 98.93 | 98.97 | 98.98 | 99.00 | 99.02 |
>
> **Result of OfficeHome dataset.**
>
> Figure: https://user-images.githubusercontent.com/17319095/201453374-414a1756-2177-4d5a-90cb-baa31ccdebb9.png
>
> | Model pool size |   1   | Previous _SOTA_ |   2   |   10  |   30  |   50  |  100  |  150  |  200  |  250  |  283  |
> |:-----:|:-----:|:-----:|:-----:|:-----:|:-----:|:-----:|:-----:|:-----:|:-----:|:-----:|:-----:|
> | SIMPLE          | 78.2 | _83.9_ | 84.94 | 85.72 | 85.93 | 88.26 | 88.17 | 88.05 | 87.94 | 87.87 | 87.74 |
>
> This experiment has been added in Appendix A.12.
>
> In summary,
>
> (1) Building a large model pool is *necessary* and *inexpensive*: only such a large model pool with diverse models allows us to match well-performing models for different distributions; and obtaining these pretrained models is of low cost.
>
> (2) The philosophy of optimizing the model pool size is embedded in the original design of SIMPLE, which allows us to **use only a limited number of models for each domain/sample**.
>
> Admittedly, the direction of optimizing the model pool size is interesting, which we leave as future work.
>
> ---

---

> > ### Comment · Reviewer_ySiZ · 2022-11-17
> > **Upgrade the paper score to 5**
> >
> > Thank you for your response. Most of my questions have been answered, then I decided to upgrade the score of the paper to 5.

---

> > > ### Author Response · Authors · 2022-11-18
> > > **Thank you for your reply!**
> > >
> > > Dear Reviewer ySiZ,
> > >
> > > We sincerely thank you for your kind reply and the concrete suggestions on our paper. Your suggestions and comments have greatly helped polish our paper, from the views of the theorem (Q1, Q2, Q3 and Q4) and the practical usage of our method (Q5).
> > >
> > > We are also open to further comments and suggestions for improving our paper.
> > >
> > > Best wishes,
> > >
> > > The authors.

---

> ### Author Response · Authors · 2022-11-14
> **Response to Reviewer ySiZ (part 1)**
>
> We are greatly thankful to the reviewer for the comments. We hope the following replies resolve the concerns raised in the review.
>
> > **Q1. (mentioned in `[Weakness]`) "could not find the role of pre-trained models in the generalization error in a DG"**
>
> The information of the pretrained model (its network architecture and weights) is in the kernel spectrum $\mathbf{\Lambda}$ and kernel basis, which influences the second term of generalization error in Theorem 1:
> $E_g = E^{\text{matched}}_g + \kappa \bar{\mathbf{a}}^T (P\mathbf{\Lambda} + \kappa \mathbf{I})^{-1}\mathscr{O}'(P\mathbf{\Lambda} + \kappa \mathbf{I})^{-1}\bar{\mathbf{a}}.$
>
> And the value of the matrix $\mathscr{O}'$ depends on the alignment between the kernel basis and the test distribution.
> If these two are matched well, the eigenvalues of $\mathscr{O}'$ may be negative and then $E_g$ may be even better than the in-distribution error $E^{\text{matched}}_g.$
>
> In conclusion, different pretrained models are suitable for different distribution shifts.
> Thus, our method SIMPLE incorporates diverse model pool with different network architectures and model weights, to provide extensive potentially well-matched models for different distributions.
>
> Thanks for pointing this out.
> We have provided more clarification on this point in section 2.2.
>
> ---
>
> > **Q2. (mentioned in `[Weakness]`) "The idea of (pre-trained) model-sample matching in the proposed method may be impractical since in DG we are not allowed to have access to the samples in the target domain."**
>
> We agree with the reviewer that the computation of OOD generalization bound in the theorem requires the access to the target distribution. However, *in our method, we do not need to access the samples in the target domain to train the model*.
>
> During training, we mimic the distribution shifts for the pretrained models, since the source domains (for training) are unseen for the pretrained models from the pretraining domain.
> Thus, training our method (i.e., the dispatcher network and the label adapter) on source domains can be regarded as handling distribution shifts in unseen domains and our dispatcher learns to match appropriate models for different OOD samples.
> Then, it will be directly evaluated on the unseen target domain.
>
> Thus, our method does not need to access the samples in the target domain during training.
>
> **Clarification of pretraining domain, source domain, and target domain.**
> We regard the data distribution used by pretrained models as *"pretraining domain"*, and the training of our method is on the source domains following the general settings in domain generalization task, which is unseen for the pretrained models since we keep all the pretrained models fixed. And the evaluation is on target domains, which are unseen during the training process.
>
> ---
>
> > **Q3. (mentioned in `Clarity`) "The term $\mathbf{a}$ is undefined in that theorem."**
>
> Thanks for pointing this, $\mathbf{a}$ is the coefficients of the target function in the Mercer basis.
> We have included this description of $\mathbf{a}$ in the detailed version of the theorem in Appendix A.5. We have refined the description accordingly in the paper.
>
> ---

---

### Official Review · Reviewer_qJrT · 2022-10-24

**Confidence:** 4
**Correctness:** 3
**Technical Novelty And Significance:** 3
**Empirical Novelty And Significance:** 3
**Recommendation:** 5

**Clarity, Quality, Novelty And Reproducibility:**

Clarity: Most parts of the paper is clear and easy to follow except for some details of the method. See the weakness above.

Quality: The paper investigates sufficient pretrained models for domain generalization, which provide good insight and motivation for the proposed method. But I still have some concerns about the method as in the weakness above.

Novelty: The paper is somewhat novel in that it learns sample specific ensemble of the pretrained models to handle distribution shifts.

Reproducibility: The paper provides enough details for reproduction. But run such many of pretrained models is not easy.


**Strength And Weaknesses:**

[+] Instead of fine-tuning pretrained models on source domains, the paper proposes to learn to dispatch pretrained models by matching the models to each test sample, which is efficient and interesting.

[+] The proposed method achieves significant performance improvement compared with the other state-of-the-arts. The ablation studies also demonstrate the effeciveness of the method.

[-] The method tends to select the best matching pretrained models for domain generalization. However, it is not sure whether method works because it handles the domain shifts or due to the strong capability of the pretrained models. For instance, the CLIP model in Model Pool B is trained on a large dataset, which contains a huge amount of different domains.

[-] Since the method requires to process the inputs by all models in the Model Pool, there should be a big computations and memory usage cost in both training and test stages. While it is shown in Table 2 that the method achieves low FLOPs as shown. Can the authors explain this? The GPU memory usage is also important for reproducing the method.

[-] The method trains a shared label space adapter using Model Pool A. However, it is not clear how to train the adapter with Model Pool B, which contains pretrained models with different sizes of output spaces. If different adapters are used for different models, will the training procedure be influenced by the ensemble weights? For example, some of the adapters will not be trained well since the weights are small.

[-] The method learns model embeddings for the model-sample matching, but there is no information of the models considered in the embeddings. How can we make sure the embeddings related to different models?


**Summary Of The Paper:**

The paper investigates performance of different pretrained models on domain generalization and shows that there is no single best pretrained model that generalizes across all distribution shifts. Based on the finds, the authors propose a new method to learn to dispatch the best matched models for each out-of-distribution sample and predict with ensemble of the models. The method does not need to fine-tune the pretrained models and achieves obvious performance improvements.

**Summary Of The Review:**

The paper provides an interesting investigation of different pretrained models on domain generalization, which well motivates to learn to dispatch different models for different target samples. The idea is interesting, but there are still some unclear parts in the methods and experiments that need to be clarified as mentioned in the weaknesses. I would consider to raise my score if the authors could well explain or address the weaknesses.

---

> ### Author Response · Authors · 2022-11-14
> **Response to Reviewer qJrT (part 2)**
>
> > **Q3. "If different adapters are used for different models, will the training procedure be influenced by the ensemble weights? For example, some of the adapters will not be trained well since the weights are small."**
>
> Yes, the training process of label space adapters for each model will be influenced by the ensemble weights.
> However, in our paper, we train the (same) adapter for all the pretrained models from the same pretraining domain, which is more efficient (than training different adapters for different pretrained models) and also avoids unexpected training instability.
> When considering the large model pool with various models from different pretraining domains (e.g., model pool B), the influence will have little negative impact or even be beneficial since
> 1. adapters are trained on all the training samples, so their training is sufficient;
> 2. since the number of parameters of adapters is relatively small, the number of training samples for adapters will not have a significant impact on the final performance;
> 3. although the training of adapters is influenced by the ensemble weights, it is also optimized towards the final objective which may not have negative impacts.
>
> #### **New Experiment**
>
> Furthermore, we conduct new experiments to verify *whether training label space adapters under the influence of ensemble weights will affect the training of adapters or degrade the performance*.
>
> **Experimental setup:**
> We compare single model performance with and without the influence of ensemble weights on the corresponding label space adapter.
> For *no influence on adapters*, we separately train individual adapters for each model.
> We select several models which are assigned with small ensemble weights in our method SIMPLE as that in your example.
> The evaluation is on *"art"* domain of the OfficeHome dataset.
>
> **Experimental results:**
> The following figure shows the performance difference between (1) a model with the adapter trained with and without ensemble weights (blue bars); and (2) our method SIMPLE and the SOTA baseline for reference (orange bar):
> https://user-images.githubusercontent.com/17319095/201605250-4787e58b-16b5-4934-8ef8-36d8a7736018.png
>
> As can be seen, \
> (1) single model performance does not significantly degrade under the influence of ensemble weights (even less than the difference between our method SIMPLE and the existing SOTA).\
> (2) Moreover, for some models in the figure, training a shared adapter (i.e., influenced by the ensemble weights) can make a single model perform even better.
> The possible reason is that training a shared adapter for multiple models avoids overfitting, leading to better generalization.
>
> Thanks for pointing this out, we have added this experiment in Appendix A.9.3.
>
> ---
>
> > **Q4. "There is no information of the models considered in the embeddings. How can we make sure the embeddings related to different models?"**
>
> Just like learning word embedding via word-word co-occurrence statistics [4], our method SIMPLE learns model information through capturing model-sample relations.
> The difference is that, word embedding [4] models the discrete multi-hot word relations, while SIMPLE models smoothed and continuous model-sample relations, i.e., the matching metric among the pretrained models and the unseen sample.
> Therefore, our embedding certainly learns information about models, and the empirical results have shown it is quite effective.
>
> But if more side information of the pretrained models can be incorporated, such as the features of the model architecture or parameter number, SIMPLE may be more effective in performance and more flexible for newly added pretrained models.
> Thank you for your constructive comments!
> We would consider it as a future exploration.
>
> ---
>
> References:\
> [1] Cha J, Lee K, Park S, et al. Domain Generalization by Mutual-Information Regularization with Pre-trained Models. ECCV 2022.\
> [2] Radford, Alec, et al. "Learning transferable visual models from natural language supervision." International Conference on Machine Learning. PMLR, 2021.\
> [3] Kumar, Ananya, et al. "Fine-tuning can distort pretrained features and underperform out-of-distribution." ICLR 2022.\
> [4] Mikolov, Tomas, et al. "Efficient estimation of word representations in vector space." arXiv preprint arXiv:1301.3781 (2013).

---

> ### Author Response · Authors · 2022-11-14
> **Response to Reviewer qJrT (part 1)**
>
> Thanks for your constructive suggestions and comments, which are very helpful for us to improve this paper. Next, we would like to respond to each of the concerns raised in your comments.
>
> > **Q1. "It is not sure whether method works because it handles the domain shifts or due to the strong capability of the pretrained models"**
>
> Our method tackles domain shifts by leveraging the diverse capability of the pretrained models in OOD generalization, in a more effective (by matching suitable models for various distributions) and efficient (without fine-tuning any pretrained models) way.
>
> Our theoretical analysis in Section 2.2 and the empirical findings in Section 2.1 illustrate that\
> (a) There is *no* dominant pretrained model over *all* the unseen domains,\
> (b) Though pretrained models may possess decent generalization ability for *some* OOD samples;\
> Thus, handling domain shifts is very important even with strong pretrained models.\
> So we propose to model the matching between pretrained models and unseen domains, at a meta-level.
> During training, we mimic the distribution shifts for the pretrained models, since the source domains (for training) are unseen for the pretrained models from the pretraining domain, as explained in Q1 of the general response part.
>
> Some recent works for the domain generalization task also realized similar findings while implicitly leveraging the pretrained model capacity for improving generalization.
> For example, the current SOTA method MIRO [1] utilizes models pretrained on ImageNet and other large-scale datasets (e.g., CLIP [2]).
>
> However, they leverage the pretrained models through fine-tuning the model parameter on the source domains and testing on the target domains, which may destruct the generalization ability compared to linear probing as shown in [3].
> While we make one step further than linear probing and keep the pretrained models fixed and directly model the matching metric between pretrained model and unseen distributions, which is more efficient in performance and effective in training.
>
> ---
>
> > **Q2. "There should be a big computations and memory usage cost in both training and test stages since the method requires to process the inputs by all models in the Model Pool."**
>
> Since (1) all the pretrained models in the model pool are fixed and (2) only a small fraction of the models would be executed, our method SIMPLE does not involve large computation and memory usage cost. We explain in detail from two perspectives below.
>
> In the *training* process, we only need to perform forward computation of each model only ONCE, to obtain the prediction outputs upon the training set, and save the results for further training procedure. Thus, the pretrained models do not need to store in memory. Additionally, the learnable parameters in our method (as shown in Figure 3 of the paper) are much smaller than the models used by other domain generalization methods (as shown in Table 2 of the paper). Therefore, both the computation and memory costs can be saved.
>
> In the *inference* process, given one specific sample, the lightweight dispatcher network will first calculate the matching scores for each model and decide to select the top-$k$ matched models and activate those pretrained models for inference subsequently. Thus, note that, not all the models will be executed. And we find that selecting top-$k \geq 2$ for inference is enough to beat the SOTA baseline as shown in Figure 6 of Appendix 9.1, which is efficient for inference.
> Moreover, since the inference of different pretrained models is isolated, the execution of the activated pretrained models could also be *paralleled* in practice.
>
> The comparison of the computation cost has been detailed in Table 2 of the paper.
>
> ---

---

> ### Author Response · Authors · 2022-11-18
> **Sincerely Looking Forward to Your Suggestions**
>
> Dear Reviewer,
>
> Given that the discussion period is coming to end soon, we sincerely look forward to your reply to our response, and we are open to any discussion to improve our paper.
>
> Best wishes,
>
> The authors.

---

### Author Response · Authors · 2022-11-14
**General response**

We thank all reviewers for taking the time to review our work and for their constructive comments.

---

Here we would like to clarify a few concepts in the paper:

> **Q1: What's the relation of the pretraining domains, source domains and target domains?**

* Pretraining domains: the domains where the pretrained models have been learned.
* Source domains: the training domains in domain generalization task.
* Target domains: the test domains in domain generalization task.

"Source domains" and "target domains" follow the general domain generalization task setting.
Note that, the source domains and the target domains are all unseen for the pretrained models. Thus, training our method (i.e., the dispatcher network and the label adapter) on source domains can be regarded as handling distribution shifts in unseen domains (w.r.t. the pretraining domains).

---

We are greatly encouraged by the positive comments of reviewers, e.g.,

* The paper proposes to learn to dispatch pretrained models by matching the models to each test sample, which is efficient and interesting (Reviewer qJrT)
* The experimental results are convincing (Reviewer ySiZ)
* The authors propose a creative way how to use a pool of pretrained models to achieve better generalization on distribution shifts (Reviewer At6g)
* An approach supported by theoretical analysis and empirical evidence based on extensive evaluations for domain generalization (Reviewer KF3m)

---

We added extra explanations and experiments per the reviewers' comments. These major revisions are shown in blue font in the updated version of the main paper and the appendix. The important revisions are summarized as below:

1. *Clarifications of pretraining, source, and target domains*: We have revised descriptions in Sections 1, 2.1, and 3.1, per the comment of Reviewer At6g.
2. *More explanations of generalization error in Theorem 1*: We have provided a clearer explanation of terms in Section 2.2 and Appendix A.5, per the comment of Reviewer ySiZ.
3. *Missing related work*: We have added one related work in Appendix A.1, per the comment of Reviewer KF3m.
4. *Extra experiment 1 -- Additional baseline of "learn the averaging weights by backpropagation"*: We added experiments in Appendix A.8, per the comment of Reviewer At6g.
5. *Extra experiment 2 -- Ablation study of training label space adapters w/o the influence of ensemble weights*: We added experiments in Appendix A.9.3, per the comment of Reviewer qJrT.
6. *Extra experiment 3 -- Sensitivity analysis on the model pool size*: We added experiments in Appendix A.12, per the comment of Reviewer ySiZ.

---

### Author Response · Authors · 2022-11-17
**Looking Forward to Further Discussions**

Dear reviewers,

Thank you again for your valuable time and insightful comments!

Given that the author-reviewer discussion period will end soon, we sincerely look forward to your reply to our response to let us know if we have resolved your concerns, and we are open to any discussion to improve our paper.

Best regards!\
The authors

---

### Decision · Program_Chairs · 2023-01-20

**Decision:**

Accept: poster

**Justification For Why Not Higher Score:**

Theoretical contribution is not so significant, although they provide theoretical analysis

**Justification For Why Not Lower Score:**

It has proposed decent theoretical analysis and comprehensive empirical evaluations.


**Metareview: Summary, Strengths And Weaknesses:**

This paper address the challenging domain generalization which the data in target domain is not available. Authors have proposed a method to leverage a large number of pre-trained models to help with generalization, and solve y dispatching the pre-trained models to OOD samples based on their matching metric to the target task.

The method has been supported by theoretical analysis although theoretical contribution is not so significant.

On the other hand, it has conducted extensive empirical evaluations which demonstrated that the proposed method has better generalization performance and significantly higher training efficiency compared to existing  methods.

**Note From Pc:**

if the above contains the word "oral" or "spotlight" please see: "oral" presentation means -> notable-top-5% and "spotlight" means -> notable-top-25%. As stated in our emails, we are disassociating presentation type from AC recommendations

**Summary Of Ac-Reviewer Meeting:**

The reviewers have mainly discussed whether the some concerns raised have been address by authors sufficiently. Correspondingly some reviewers have updated their their scores as the authors addressed their concerns.